# FLOW CACHING FOR AUTOREGRESSIVE VIDEO GENERATION

**Yuexiao Ma**[1,2,†,‡], **Xuzhe Zheng**[1,†], **Jing Xu**[1,†], **Xiwei Xu**[1], **Feng Ling**[2], **Xiawu Zheng**[1], **Huafeng Kuang**[2], **Huixia Li**[2], **Xing Wang**[2], **Xuefeng Xiao**[2], **Fei Chao**[1], **Rongrong Ji**[1*]

[1]Key Laboratory of Multimedia Trusted Perception and Efficient Computing,
Ministry of Education of China, Xiamen University, 361005, P.R. China.
[2]ByteDance

## ABSTRACT

Autoregressive models, often built on Transformer architectures, represent a powerful paradigm for generating ultra-long videos by synthesizing content in sequential chunks. However, this sequential generation process is notoriously slow. While caching strategies have proven effective for accelerating traditional video diffusion models, existing methods assume uniform denoising across all frames—an assumption that breaks down in autoregressive models where different video chunks exhibit varying similarity patterns at identical timesteps. In this paper, we present **FlowCache**, the first caching framework specifically designed for autoregressive video generation. Our key insight is that each video chunk should maintain independent caching policies, allowing fine-grained control over which chunks require recomputation at each timestep. We introduce a chunkwise caching strategy that dynamically adapts to the unique denoising characteristics of each chunk, complemented by a joint importance–redundancy optimized KV cache compression mechanism that maintains fixed memory bounds while preserving generation quality. Our method achieves remarkable speedups of **2.38**× on MAGI-1 and **6.7**× on SkyReels-V2, with negligible quality degradation (VBench: $0.87 \uparrow$ and $0.79 \downarrow$ respectively). These results demonstrate that FlowCache, successfully unlocks the potential of autoregressive models for real-time, ultra-long video generation—establishing a new benchmark for efficient video synthesis at scale. The code is available at https://github.com/mikeallen39/FlowCache.

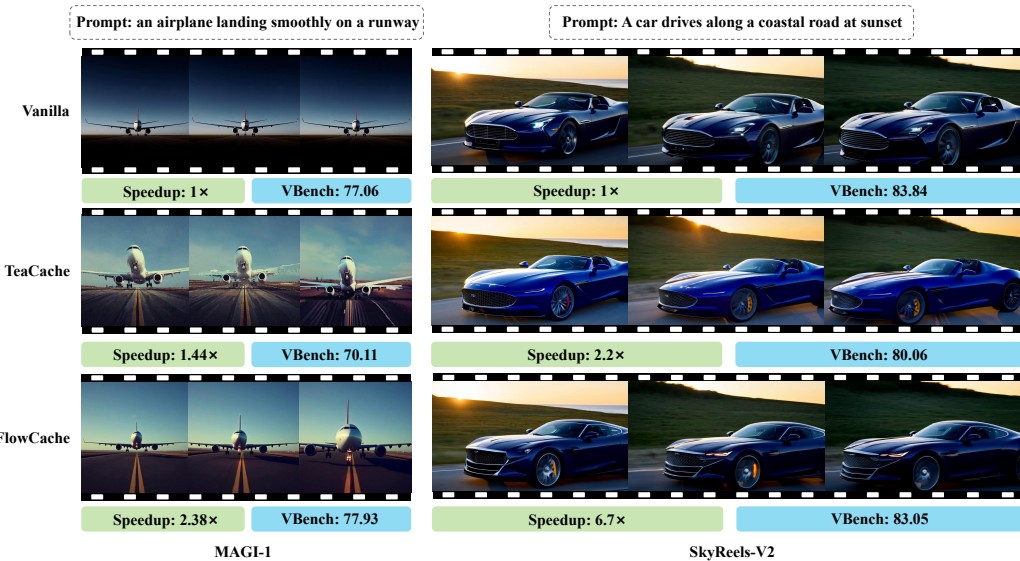

Figure 1: Visual comparison of video generation quality among the vanilla model, TeaCache, and our FlowCache.

---

*Corresponding author: rrji@xmu.edu.cn, †: Equal contribution, ‡: This work was done when Yuexiao Ma was intern at ByteDance.

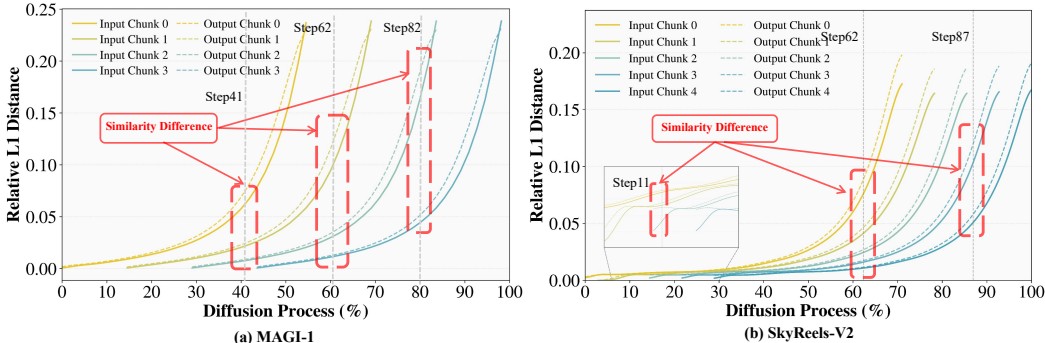

Figure 2: Adjacent-timestep relative L1 distance across denoising trajectories for autoregressive video diffusion models. Denoising progress (%) and relative L1 distance are shown on horizontal and vertical axes, respectively, with colors denoting distinct video chunks. Results for (a) MAGI-1 and (b) SkyReels-V2 reveal three patterns: (i) monotonic increase in relative L1 distance during denoising, confirming Theorem 1; (ii) substantial metrics divergence among chunks at identical timesteps reflects heterogeneous denoising stages, yielding variable reuse probabilities; (iii) persistently high similarity between model inputs and sampler outputs across all chunks.

# 1 INTRODUCTION

While large language models have demonstrated remarkable reasoning and task execution capabilities (Zhang et al., 2024a; 2025b; Zhou et al., 2025; Wu et al., 2024; Ji et al., 2024; Wang et al., 2025; Zhang et al., 2025a; Yan et al., 2025a;b; Ma et al., 2025; Li et al., 2025e; Lin et al., 2026), video generation models (Lin et al., 2024a; Zheng et al., 2024; Kong et al., 2024; Peng et al., 2025; Wan et al., 2025; Brooks et al., 2024; Gao et al., 2025) have demonstrated the ability to capture the objective physical laws governing the real world and are increasingly applied across a wide range of domains, including autonomous driving (Fu et al., 2024; Ma et al., 2024a), social media content creation (Brooks et al., 2024), special effects generation (Gao et al., 2025), and personalized product demonstrations (Wan et al., 2025), supported by theoretical advancements in generation dynamics (Sun et al., 2025; Li et al., 2025a; Zhong et al., 2025b; Wu et al., 2025). Recent works have also explored digital human synthesis (Li et al., 2025b) and efficient scaling strategies for high-quality video generation (Cheng et al., 2025; Cui et al., 2025; 2026; Yuan et al., 2025; Lu et al., 2025). Among these models, the diffusion transformer architecture (DiT) Peebles & Xie (2023) has emerged as a mainstream approach due to its superior generative quality compared to traditional UNet-based structures (Ronneberger et al., 2015), largely attributed to its strong scalability. However, the generation of long videos imposes prohibitively high computational costs, which significantly restricts the practical applicability of these models and limits the potential for personalized customization. In particular, the computational complexity of DiT inference scales quadratically with the length of the video token sequence, as a result of the transformer's requirement to compute attention scores between all pairs of tokens. Moreover, this challenge is further exacerbated by the multi-timestep denoising process applied across video frames.

To decouple inference computational costs from sequence length, autoregressive video generation models (Teng et al., 2025; Chen et al., 2025a) based on Causal Diffusion-Forcing (CDF) (Chen et al., 2024a; Yin et al., 2025; Song et al., 2025) inference paradigms are garnering increasing attention. These models achieve this by partitioning the entire video sequence into fixed-frame video chunks and performing causal autoregressive denoising on them. With a fixed denoising window size, the computational complexity of long-video inference drops from quadratic to linear scaling, thereby establishing theoretical feasibility for real-time long-video generation. Nevertheless, at high resolutions, the computational demands of autoregressive video models for producing extended videos remain considerable, creating a substantial gap between current capabilities and the goal of real-time generation. For example, using an $A800$ GPU with batch size 1, generating a 10-second video at $720 \times 720$ resolution with full KV cache reference requires approximately 32GB of memory and 50 minutes of inference time for the MAGI-1-4.5B-distill model.

To address these challenges, a class of training-free, cache-based methods (Yuan et al., 2024; Liu et al., 2025b;a; Kahatapitiya et al., 2024; Bu et al., 2025; Chen et al., 2024c; Zhao et al.; Ma et al., 2024b; Selvaraju et al., 2024) has been developed to accelerate the denoising process in DiT models. These approaches are orthogonal to various compression techniques, such as quantization (Qin et al., 2020; Huang et al., 2024a; Qin et al., 2024; 2022; Ashkboos et al., 2024; Frantar et al., 2022; Li et al., 2024a; Zhong et al., 2023; 2025a; 2024; Lin et al., 2024b; 2025; Shao et al., 2023; Chen et al., 2025b; 2024b; 2025d;c; He et al., 2023b;a; Yang et al., 2025a) and sparsification (Xu et al., 2025; Xi et al., 2025; Yang et al., 2025b; JAISWAL et al.; Liu et al., 2021b;a; 2022; Yin et al., 2023; Huang et al., 2025b;a; Zhang et al., 2023a; Lu et al., 2024). Unlike distillation-based methods (Wang et al., 2023; Yin et al., 2024; Meng et al., 2023; Sauer et al., 2024), they facilitate faster DiT inference without incurring additional training overhead. Existing caching strategies can be broadly classified into two categories based on variations in feature caching intervals: uniform caching and dynamic caching. Uniform caching (Chen et al., 2024c; Zhao et al.; Ma et al., 2024b; Selvaraju et al., 2024; Liu et al., 2025b) exploits the high similarity of model outputs across adjacent timesteps by storing and reusing them at regular intervals. In contrast, dynamic caching (Bu et al., 2025; Kahatapitiya et al., 2024; Liu et al., 2025a) employs metrics computed during inference to decide whether features at a specific timestep should be recomputed. Nevertheless, these methods are predominantly tailored to synchronous video diffusion models, where all frames undergo consistent denoising levels at the same timestep. When transitioning to autoregressive video generation models, it is challenging to achieve an optimal trade-off between acceleration ratio and video generation quality.

As illustrated in Figure 2, we present similarity metrics between adjacent timesteps for different video chunks in the MAGI-1 and SkyReels-V2 models. Across diverse autoregressive models, we identify three consistent patterns: (a) As the denoising process progresses toward the clean video, the inter-timestep similarity of video chunks progressively deteriorates. (b) Within the same timestep, different video chunks exhibit heterogeneous denoising states, leading to substantial variations in their similarity measures. (c) Throughout the entire denoising trajectory, the inputs and outputs of autoregressive models maintain high similarity across all video chunks. These observations reveal that uniform treatment of all features fails to account for the heterogeneity in denoising states across different video chunks. Specifically, at any given timestep, video chunks approaching their clean state demonstrate low temporal similarity and necessitate recomputation. In contrast, video chunks in early denoising stages exhibit high similarity to their previous timestep features, making them suitable candidates for cache reuse. This contradiction motivates our proposed **FlowCache** method, wherein each video chunk maintains an independent caching strategy to accommodate these varying computational requirements. Through extensive experimentation, we demonstrate that our approach not only achieves superior acceleration ratios but also preserves the quality of generated videos.

Furthermore, autoregressive models generate substantial key-value (KV) caches during the generation process. When conceptualizing video chunks as analogous to tokens in language models, each completed denoising iteration of a video chunk results in a proportional expansion of the KV cache. The KV cache plays a crucial role in autoregressive video generation models by enabling the modeling of physical dynamics in the objective world (Motamed et al., 2025) and controlling temporal consistency of subjects and backgrounds throughout the video sequence (Teng et al., 2025). While existing KV cache compression techniques (Wan et al., 2024; Li et al., 2024c; Zhang et al., 2023b; Cai et al., 2025) have predominantly focused on large language models, the unique characteristics of video generation present distinct challenges and opportunities. To address this gap, we present the first comprehensive investigation of KV cache compression's impact on the generative capabilities of autoregressive video generation models. Our analysis encompasses two critical dimensions: the selection of salient KV cache entries and the quantification of their relative importance. We conduct this analysis across three hierarchical granularities—key video segments, key frames, and key pixel regions—to capture the multi-scale nature of video information. Experimental results demonstrate that our proposed KV cache compression method achieves substantial computational savings while preserving the quality of generated videos, establishing its effectiveness for autoregressive video generation tasks.

In summary, our key contributions are as follows:

- We uncover fundamental heterogeneity in the denoising process of autoregressive video generation models, empirically demonstrating that different video chunks exhibit distinct similarity difference at identical timesteps. Through rigorous theoretical analysis, we es-

tablish that video chunks approaching complete denoising manifest significantly weaker similarity compared to their early-stage counterparts, providing crucial insights for optimization strategies.

- We introduce FlowCache, a pioneering method that revolutionizes caching in autoregressive video generation by treating each video chunk as an independent computational entity. This chunk-specific caching strategy dynamically adapts to the varying denoising states, maximizing both computational efficiency and generation flexibility.

- We present the first comprehensive study examining the impact of KV cache compression on autoregressive video generation models. Our multi-dimensional analysis encompasses two critical aspects—importance computation granularity and selective retention granularity—operating across three hierarchical levels: video segments, frames, and pixel regions. This systematic investigation offers valuable insights into the memory-quality tradeoffs in efficient video generation.

- Comprehensive empirical evaluation across diverse autoregressive video generation models confirms that our approach achieves superior computational efficiency while preserving generation quality. FlowCache accelerates inference by $2.38\times$ and $6.7\times$ compared to baseline methods on MAGI-1 and SkyReels-V2, respectively, with minimal impact on VBench scores ($0.87 \uparrow$ for MAGI-1, $0.79 \downarrow$ for SkyReels-V2). These results establish FlowCache as the new state-of-the-art for efficient autoregressive video generation.

## 2 RELATED WORK

**AutoRegressive Video Generation.** Our work builds upon recent advances in autoregressive video generation models (Yin et al., 2025; Teng et al., 2025; Chen et al., 2025a). These methods adapt the successful paradigms of diffusion models and large language models (LLMs) by modeling a long video sequence as an autoregressive prediction of discrete chunks, effectively overcoming the quadratic complexity challenge of long-sequence modeling. Each video chunk serves as a fundamental generation unit, analogous to a token in LLMs, with the current chunk conditioned on all prior ones. In this paradigm, the chunk generator is typically a diffusion model trained with flow matching (Lipman et al., 2022), enabling the autoregressive framework to integrate advanced diffusion-based generation for efficient production of long, high-fidelity video sequences.

**Cache Strategy.** Recent work on accelerating diffusion models—especially Diffusion Transformers (DiTs)—has centered on training-free feature caching that exploits temporal redundancy across denoising timesteps. TeaCache (Liu et al., 2025a) uses timestep embedding–modulated inputs and polynomial fitting to estimate output differences for caching decisions. ToCa (Zou et al., 2024) performs token-wise caching by assessing temporal redundancy and error sensitivity via attention-based scores. DiCache (Bu et al., 2025) dynamically aligns multi-step cache trajectories using an online shallow-layer probe to estimate output variation.

**KV Cache Compression.** Key-Value (KV) cache compression reduces memory and computation costs in large language model (LLM) inference by selectively retaining past key and value states. H2O (Zhang et al., 2023b) preserves "heavy-hitter" tokens with high cumulative attention scores. SnapKV (Li et al., 2024c) pre-selects critical tokens using a fixed observation window at the prompt's end. D2O (Wan et al., 2024) introduces a two-level discriminative framework that dynamically allocates KV cache budgets across layers and includes a compensation mechanism.Additionally, recent studies have investigated retrieval-augmented strategies and cache merging to enhance long-context comprehension in video understanding (Luo et al., 2025; 2024; Zhang et al., 2024b).

## 3 METHODOLOGY

### 3.1 PRELIMINARY

**Diffusion Model.** Diffusion models (Ronneberger et al., 2015; Peebles & Xie, 2023) achieve high-quality sample generation by establishing a reversible mapping between data and noise distributions. Their theoretical framework comprises two complementary processes: forward diffusion and reverse

denoising. The forward diffusion process progressively transforms the original data distribution $\pi_0$ into a prior distribution $\pi_1$—typically a standard Gaussian—through systematic noise injection. Within the flow matching framework (Lipman et al., 2022), this process manifests as a deterministic flow in probability space, where the intermediate state at any time $t$ is expressed as:

$$x_t = (1 - \sigma(t)) \cdot x_{data} + \sigma(t) \cdot x_{noise}. \tag{1}$$

Here, $\sigma(t)$ denotes a monotonic scheduling function that governs the smooth interpolation between data and noise domains. The reverse denoising process constitutes the core mechanism of diffusion models, wherein a learned velocity field $v_\theta$ approximates the inverse mapping from noise to data. This process is governed by the following ordinary differential equation: $\frac{dx}{dt} = v_\theta(x_t, t, c)$, where $c$ represents optional conditioning information. During inference, the model employs numerical integration schemes to solve this ODE, typically utilizing first-order Euler discretization (Karras et al., 2022) or more sophisticated sampling algorithms Lu et al. (2022); Zhao et al. (2023):

$$x_{t_{i-1}} = x_{t_i} + v_\theta(x_{t_i}, t_i, c) \Delta t_i \tag{2}$$

By sampling from the noise distribution and integrating along the learned velocity field, the model generates new samples from the target distribution. This flow matching paradigm effectively captures the intrinsic geometric structure of data manifolds through direct optimization of the velocity field prediction, thereby enabling high-fidelity synthesis.

**AutoRegressive Video Model.** Autoregressive video models (Yin et al., 2025; Teng et al., 2025; Chen et al., 2025a) circumvent the quadratic complexity of sequence length by decomposing long videos into discrete chunks, treating each as an analogous unit to language model tokens. This chunking strategy enables efficient generation of extended video sequences through autoregressive denoising. Formally, a video is partitioned into $k$ chunks, where each chunk $X_i \in \mathbb{R}^{c_{in} \times s \times h \times w}$ ($i \in \{1, \cdots, k\}$) represents a latent representation with $c_{in}$ channels, spatial dimensions $h \times w$ after downsampling, and temporal length $s$. Assuming a maximum window size of $l$ video chunks permitted for the denoising process, the denoising process for the $i$-th chunk at timestep $t$ follows:

$$X_{t-1}^i = X_t^i + v_\theta(X_t^i, t, c) \cdot \Delta t. \tag{3}$$

Where $t \in [(i-1)T/l, (i+l-1)T/l]$ represents the timestep. $v_\theta(\cdot)$ denotes the output of autoregressive diffusion model with parameters $\theta$. $c$ denotes the conditional inputs (text, image, video).

**Cache Strategy.** Various caching strategies (Bu et al., 2025; Ma et al., 2024b; Liu et al., 2025a) have been developed to accelerate diffusion model inference by storing outputs from selected timesteps for reuse in subsequent iterations. These approaches typically quantify the similarity between consecutive timesteps using importance metrics, such as the relative L1 distance:

$$L1_{rel}(X, t) = \frac{\|X_{t-1} - X_t\|_1}{\|X_t\|_1}. \tag{4}$$

Through the accumulation of these similarity measures, the system determines whether to reuse cached outputs (when the cumulative similarity remains below a predefined threshold) or to perform new model inference (when the threshold is exceeded). However, current caching strategies prove incompatible with autoregressive models due to their distinctive inference mechanisms.

### 3.2 FLOWCACHE

In the context of autoregressive video models, we calculate the relative L1 distance defined in Equation 4 between consecutive timestep for $i$-th video chunk:

$$L1_{rel}(X, t, i) = \frac{\|X_{t-1}^i - X_t^i\|_1}{\|X_t^i\|_1} = \frac{\|v_\theta(X_t^i, t, c) \cdot \Delta t\|_1}{\|X_t^i\|_1}. \tag{5}$$

The relative L1 distance metric in Equation 5 possesses inherent mathematical properties that govern the temporal dynamics of the denoising process in autoregressive video models. The theoretical framework of flow matching and the underlying structure of the diffusion process naturally lead to specific behavioral patterns in the velocity field throughout the denoising trajectory. By analyzing the mathematical relationship between the velocity predictions and the state evolution from noise to data, we can derive fundamental properties of the relative L1 distance across different time steps. The following theorem establishes a key monotonic property that emerges from the optimal velocity field under standard diffusion model assumptions:

**Theorem 1** *Assume the diffusion model $v_\theta$ has converged to the optimal velocity field of the flow matching objective. Further, assume the scheduling function is of the power-law form $\sigma(t) = (t/T)^p$ for some constant power $p > 0$ and total time $T$. Given $0 < t_1 < t_2 \leq T$ and a data chunk $X^i$ whose initial noise state $X_T^i$ is drawn from $\mathcal{N}(0, I)$, the following inequality holds:*

$$L1_{rel}(X, t_1, i) \geq L1_{rel}(X, t_2, i). \tag{6}$$

The proof appears in the Appendix B. Theorem 1 establishes that the relative L1 distance between model outputs increases as video chunks converge toward real video, reflecting diminished chunk similarity. Figure 2 visualizes this relationship by plotting the relative L1 distance between consecutive timesteps across all video chunks for various autoregressive video models, with the denoising progression on the horizontal axis and the relative L1 distance on the vertical axis. Distinct colors denote individual video chunks. Three key patterns emerge from this analysis:

- The relative L1 distance monotonically increases as denoising progresses toward real video, demonstrating systematic degradation of chunk similarity at later timesteps—a direct consequence of Theorem 1.
- Cross-sectional analysis at fixed timesteps reveals pronounced divergence among chunks (represented by different colors), indicating heterogeneous denoising stages and consequently disparate similarity metrics.
- Model inputs and post-sampler outputs exhibit consistently high similarity across all video chunks throughout the denoising trajectory.

Therefore, based on Theorem 1, we derive the following corollary:

**Corollary 1** *Assuming that distinct video chunks $i \neq j$ represent heterogeneous content such that their state norms differ at any intermediate timestep $t \in (0, T)$ (i.e., $\|X_t^i\|_1 \neq \|X_t^j\|_1$), and that the model's update magnitude $\|v_\theta(X_t^k, t, c) \cdot \Delta t\|_1$ is approximately invariant across chunks for a fixed $t$, then their relative L1 distances are unequal:*

$$L1_{rel}(X, t, i) \neq L1_{rel}(X, t, j). \tag{7}$$

The proof appears in the Appendix C. The preceding theorem and corollary establish the necessity of maintaining independent cache strategies for individual chunks. We therefore introduce FlowCache, a novel caching methodology tailored for autoregressive video models.

Figure 3 illustrates that conventional caching approaches employ uniform strategies across all video chunks, failing to accommodate the heterogeneous similarity characteristics arising from disparate denoising stages within the same timestep. This uniformity constrains flexibility, limits acceleration potential, and ultimately degrades generation quality. In contrast, FlowCache independently evaluates each chunk's similarity profile. Specifically, for the $i$-th video chunk:

$$f(X, t, i) = \begin{cases} 0 & \text{if } t \in (T - m, T], \\ 0 & \text{if } t \in [0, T - m] \text{ and } f(X, t + 1, i) + L1_{\text{rel}}(X, t, i) > \epsilon, \\ f(X, t + 1, i) + L1_{\text{rel}}(X, t, i) & \text{if } t \in [0, T - m] \text{ and } f(X, t + 1, i) + L1_{\text{rel}}(X, t, i) \leq \epsilon. \end{cases} \tag{8}$$

where $\epsilon$ denotes the threshold value and $m$ represents the number of initial timesteps excluded from cache reuse ($m = 5$ for MAGI-1, $m = 4$ for SkyReels-V2). When $f(X, t, i) = 0$, forward computation is performed for $i$-th video chunk; otherwise, cached activations are reused. Our empirical analysis demonstrates that excluding early timesteps is critical for preserving generation quality. The threshold values are: MAGI-slow: 0.01, MAGI-fast: 0.015, SkyReels-V2-slow: 0.1, SkyReels-V2-fast: 0.15.

Theorem 1 demonstrates that any chunks approaching full denoising exhibit diminished similarity at given timestep, warranting reduced reuse. Conversely, chunks proximate to Gaussian noise maintain higher similarity, enabling reuse across multiple consecutive timesteps. By fully considering the inference paradigm of autoregressive models, FlowCache maximizes caching flexibility and achieves substantial performance gains. Empirically, our method delivers $2.38\times$ acceleration on MAGI-1 with $0.87\%$ improvement in VBench scores relative to base model, and $6.7\times$ speedup on SkyReelsV2 with $0.79\%$ VBench score degradation.

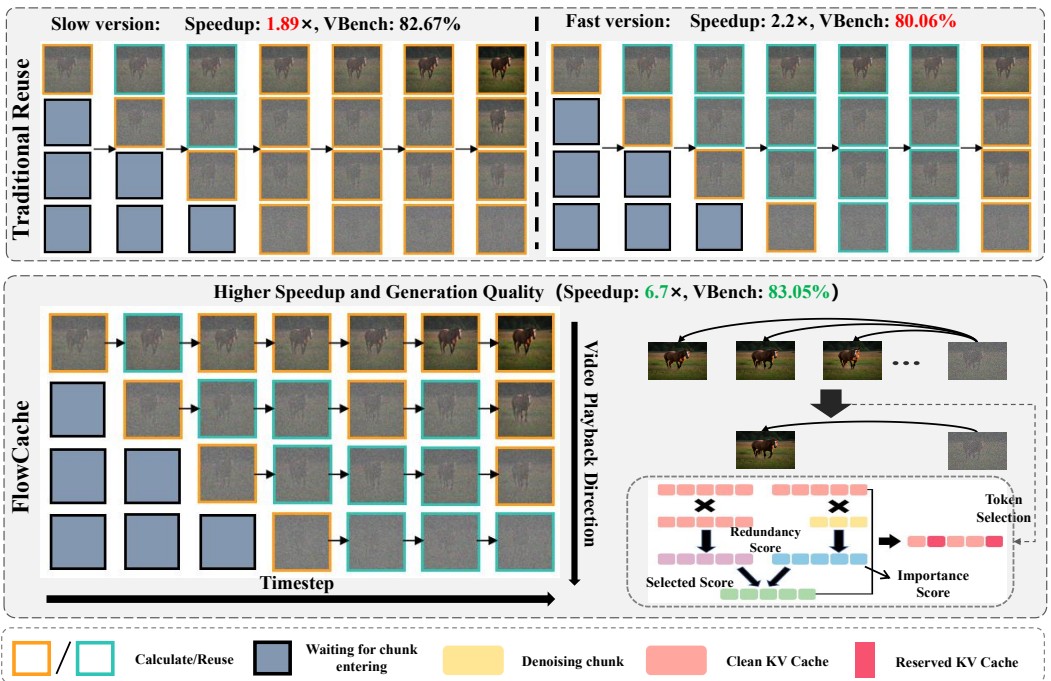

Figure 3: Comparison of caching strategies in autoregressive video generation. The top part illustrates the Traditional Reuse strategy, which applies a uniform caching policy across all video chunks (i.e., all chunks at the same timestep share the same compute/reuse status, except for the newly initialized denoising chunk, which must be computed). In contrast, our FlowCache (bottom left) employs a chunkwise adaptive caching policy, dynamically deciding for each chunk whether to reuse cached features or perform recomputation based on its own relative L1 distance trajectory. The bottom right panel details FlowCache's KV cache management: it maintains a Reserved KV Cache that selectively retains historically important and non-redundant tokens.

## 3.3 KVCACHE COMPRESSION

Autoregressive video generation models synthesize videos of arbitrary length by sequentially denoising groups of spatiotemporal chunks—mirroring the autoregressive token generation process in large language models (LLMs). In this paradigm, fully denoised video chunks (i.e., clean video chunks) are stored in a Key-Value (KV) cache, which is attended to by the group of chunks currently undergoing denoising. This mechanism enables the model to condition on historical context and maintain temporal coherence throughout the sequence.

In LLMs, KV cache compression commonly relies on importance scoring: historical tokens are ranked by their attention relevance to recent context, and only the top entries are retained (e.g., H2O (Zhang et al., 2023b), D2O (Wan et al., 2024)). While effective for text, this strategy performs poorly in video generation under a fixed cache budget. The core issue stems from the high spatial and temporal redundancy inherent in video data: many tokens assigned high importance scores are nearly identical. As a result, importance-only selection retains multiple redundant yet individually salient segments. This inefficient allocation starves the cache of diverse, potentially useful historical content, ultimately degrading temporal consistency.

We argue that effective KV cache compression for video generation must jointly account for **importance** and **redundancy**—a principle also adopted in efficient reasoning models (e.g., R-KV (Cai et al., 2025)). Our approach selects historical entries that are both relevant to the current denoising tokens and mutually dissimilar, ensuring optimal use of the limited cache budget.

Specifically, we allocate a fixed-size KV cache buffer $B_{\text{total}}$, partitioned into two regions:

1. A **compressed clean-chunk region** of size $B_{\text{budget}}$, storing compressed KV states from all clean chunks.

2. A **current denoising region** of size $B_{\text{active}}$, holding the KV states of the group of video chunks currently undergoing denoising.

When $B_{\text{total}}$ is full, we concatenate the KV cache from the clean-chunk region (uncompressed during the first compression step) with the KV states of any chunks in the current region that have just completed denoising. The merged set is then compressed via our selection criterion. After compression, the current denoising region is partially freed, creating space for the next undenoised chunk.

Similar to R-KV (Cai et al., 2025), our selection criterion balances importance and redundancy. Let $Q \in \mathbb{R}^{L_q \times H_q \times d}$ denote the queries from the current denoising tokens, and $K_{\text{clean}} \in \mathbb{R}^{L_k \times H_k \times d}$ the keys from clean chunks, where $L_q$, $L_k$ are sequence lengths, $H_q$, $H_k$ are numbers of query and key heads, and $d$ is the head dimension. The importance of each historical token is derived from the attention weights between $Q$ and $K_{\text{clean}}$. Specifically, we compute attention scores, apply softmax over the key dimension, and average across the query sequence dimension to obtain a per-head importance distribution over historical tokens:

$$Imp_j^{(h)} = \frac{1}{L_q} \sum_{i=1}^{L_q} \left[ \text{softmax}\left( \frac{q_i^{(h)}(K_{\text{clean}}^{(h)})^\top}{\sqrt{d}} \right) e_j \right], \tag{9}$$

where $h$ indexes the key head (with query heads grouped accordingly), and $q_i^{(h)}$, $k_j^{(h)}$ are the corresponding query and key vectors. The vector $e_j \in \mathbb{R}^{L_k \times 1}$ is a one-hot column vector containing a "1" at the j-th position and zeros elsewhere. To enhance robustness, we apply a 1D max-pooling with kernel size $k$ and padding $k/2$ to $\text{Imp}^{(h)}$:

$$\widetilde{\text{Imp}}^{(h)} = \text{MaxPool1D}\left( \text{Imp}^{(h)}; \text{kernel} = k, \text{padding} = k/2 \right). \tag{10}$$

For redundancy, we compute per-head cosine similarity matrices over $K_{\text{clean}}$. After $\ell_2$-normalizing the keys, we form $S_{ij}^{(h)} = (k_i^{(h)})^\top k_j^{(h)}$ for each head $h$, explicitly set diagonal entries to zero ($S_{ii}^{(h)} = 0$), and average across the token dimension and apply softmax:

$$\text{Red}_j^{(h)} = \text{softmax}\left( \frac{1}{L_k} \sum_{i=1}^{L_k} S_{ij}^{(h)} \right)_j. \tag{11}$$

This yields a per-head redundancy distribution $\text{Red}^{(h)} \in \mathbb{R}^{L_k}$, where higher values indicate tokens that are, on average, more similar to others in the same head.

Finally, we combine the pooled importance and redundancy into a unified per-head selection score:

$$\text{Score}_j^{(h)} = \lambda \cdot \widetilde{\text{Imp}}_j^{(h)} - (1 - \lambda) \cdot \text{Red}_j^{(h)}, \tag{12}$$

where $\lambda \in [0, 1]$ is a mixing coefficient. For each attention head $h$, we select the top-$B$ tokens with the highest $\text{Score}^{(h)}$, where $B$ is the compression budget.

By jointly optimizing for relevance and diversity, our method preserves long-range temporal consistency while substantially reducing the memory and computational overhead of DiT attention—enabling efficient, high-fidelity long-form video generation.

Furthermore, we provide an analysis of the efficiency of KV cache compression—covering both GPU memory footprint and computational overhead. For details, please see Appendix D.2.

## 4 EXPERIMENTS

### 4.1 SETTINGS

**Base Models** To evaluate the efficacy of our proposed method, we selected two representative diffusion models based on the autoregressive paradigm: MAGI-1-4.5B-distill (Teng et al., 2025) and

Table 1: Quantitative evaluation of inference efficiency and visual quality in autoregressive video generation models.

| Model | Method | PFLOPs ↓ | Speedup ↑ | Latency(s) ↓ | VBench ↑ | LPIPS ↓ | SSIM ↑ | PSNR ↑ |
|---|---|---|---|---|---|---|---|---|
| MAGI-1 | Vanilla | 306 | 1× | 2873 | 77.06% | - | - | - |
| | TeaCache-slow | 294 | 1.12× | 2579 | 77.50% | 0.6211 | 0.2801 | 13.26 |
| | TeaCache-fast | 225 | 1.44× | 1998 | 70.11% | 0.8160 | 0.1138 | 8.94 |
| | FlowCache-slow | 161 | 1.86× | 1546 | **78.96%** | **0.3160** | **0.6497** | **22.34** |
| | FlowCache-fast | **140** | **2.38×** | **1209** | 77.93% | 0.4311 | 0.5140 | 19.27 |
| SkyReels-V2 | Vanilla | 113 | 1× | 1540 | 83.84% | - | - | - |
| | TeaCache-slow | 58 | 1.89× | 814 | 82.67% | 0.1472 | 0.7501 | 21.96 |
| | TeaCache-fast | 49 | 2.2× | 686 | 80.06% | 0.3063 | 0.6121 | 18.39 |
| | FlowCache-slow | 36 | 5.88× | 262 | **83.12%** | **0.1225** | **0.789** | **23.74** |
| | FlowCache-fast | 28 | **6.7×** | **230** | 83.05% | 0.1467 | 0.7635 | 22.95 |

SkyReels-V2-1.3B-540P (Chen et al., 2025a). Although both models inference autoregressively, their implementations differ significantly. MAGI-1 processes a video by dividing it into chunks, applying autoregressive modeling between these chunks. A sliding window mechanism is employed to constrain the number of chunks being denoised simultaneously. In contrast, SkyReels-V2 adopts a hierarchical approach: each video chunk is further partitioned into blocks. Autoregression occurs within a chunk, where earlier blocks are denoised for a predefined number of steps before subsequent blocks begin the denoising process. This design results in non-uniform denoising levels across blocks at any given denoising step.

**Evaluation Metrics** A comprehensive evaluation is conducted, focusing on two critical aspects: the perceptual quality of the generated videos and the computational efficiency during inference. For quality assessment, we adopt established metrics including VBench (Huang et al., 2024b), LPIPS (Zhang et al., 2018), PSNR, and SSIM, following TeaCache (Liu et al., 2025a). We also acknowledge recent advancements in blind and reference-based image quality assessment which focus on semantic and distortion sensitivity (Li et al., 2023; 2024b; 2025f;d;c). To quantify efficiency, we measure the computational cost in Floating Point Operations (FLOPs) and the practical inference latency. For all video generation quality assessments, we utilize the VBench-long benchmark, which we refer to as VBench throughout the paper for brevity.

**Implementation Detail** All experiments are implemented using PyTorch and executed on NVIDIA A800 80GB GPUs. For MAGI-1, each chunk consists of 24 frames, denoised for 64 steps per chunk, with a total of 10 chunks generated. For SkyReels-V2, the inference configuration uses chunks of 97 frames with an overlap of 17 frames between consecutive chunks. Each block within a chunk is denoised for 50 steps, and a total of two chunks are generated. Additionally, the execution of KV cache compression and the measurement of speedup ratios are explained in the Appendix D.1.

## 4.2 RESULTS

Quantitative results in Table 1 demonstrate the clear superiority of FlowCache over TeaCache. Evaluated under both slow and fast configurations, FlowCache delivers higher video quality and reduced latency across diverse models and acceleration ratios. As shown in Table 1, FlowCache demonstrates superior efficiency-quality trade-offs compared to TeaCache. On MAGI-1, while TeaCache-fast suffers a significant quality degradation (VBench drops from 77.50 to 70.11) when accelerating from 1.12× to 1.44×, FlowCache-fast achieves a 2.38× speedup while maintaining high visual quality (VBench 77.93), even slightly surpassing the baseline. FlowCache-slow delivers the best quality among all variants with a 1.86× acceleration.The advantage is more pronounced on SkyReels-V2. FlowCache-slow achieves 5.88× acceleration with minimal quality loss (VBench 83.12), significantly outperforming TeaCache-slow (1.89×, VBench 82.67). FlowCache-fast maintains excellent quality (VBench 83.05) at 6.7× speedup, whereas TeaCache-fast drops to 80.06 at 2.2× acceleration.

Table 2: Ablation Study of FlowCache's Key Components: Reuse Strategy and KV Cache Compression.

| Model | Reuse Strategy | KV Cache Compression | VBench ↑ | LPIPS ↓ | SSIM ↑ | PSNR ↑ |
|---|---|---|---|---|---|---|
| MAGI-1 | - | - | 77.06% | - | - | - |
| | TeaCache | - | 70.11% | 0.8160 | 0.1138 | 8.94 |
| | ChunkWise | - | 77.66% | **0.4208** | **0.5226** | **19.89** |
| | ChunkWise | Enabled | **77.93%** | 0.4311 | 0.5140 | 19.27 |
| SkyReels-V2 | - | - | 83.34% | - | - | - |
| | TeaCache | - | 80.06% | 0.3063 | 0.6121 | 18.39 |
| | ChunkWise | - | **83.12%** | **0.1225** | **0.789** | **23.74** |
| | ChunkWise | Enabled | 83.01% | 0.1565 | 0.7425 | 22.61 |

## 4.3 ABLATION STUDY

**Effectiveness of Individual Components.** We conduct ablation studies on two key components—reuse strategy and KV cache compression—to validate the effectiveness of our approach. To ensure a fair comparison, we consistently employ the fast variant across all ablation settings. As shown in Table 2, on the MAGI-1 model, applying TeaCache (Liu et al., 2025a) reuse strategy leads to a significant degradation in video generation quality, whereas chunkwise reuse preserves the model's original performance. Moreover, KV cache compression has a negligible impact on generation quality, confirming the soundness of our designed KV cache compression mechanism. Consistent improvements are observed on the SkyReels-V2 model. TeaCache reuse strategy results in substantial quality degradation. In contrast, chunkwise reuse achieves acceleration comparable to TeaCache but without compromising performance, robustly demonstrating its effectiveness. The incorporation of KV cache compression again incurs only a negligible performance loss, further validating the generalizability of our method.

**Application on the 16-step distilled model.** We conducted comprehensive experiments on the 16-step distilled MAGI-1 model. Due to computational constraints, we evaluated performance using representative VBench metrics selected based on established practices in video generation compression research (Zhao et al., 2024; Feng et al., 2025; Yang et al.; Shao et al., 2025). To facilitate an intuitive comparison, we compute the average scores of the selected VBench metrics using the official normalization and weighting methodology provided by the VBench benchmark. For details, please see Appendix E.1.

## 5 CONCLUSION

In this work, we have presented FlowCache, a novel and efficient framework specifically designed to accelerate autoregressive video generation models. Our approach is grounded in a key empirical and theoretical insight: the denoising trajectories of individual video chunks are highly heterogeneous, exhibiting distinct levels of temporal similarity at any given timestep. To address this, FlowCache introduces a chunkwise adaptive caching strategy that independently manages the reuse policy for each video chunk based on its unique denoising state. Furthermore, to tackle the substantial memory overhead from the growing Key-Value (KV) cache—a mechanism crucial for maintaining long-range temporal consistency—we propose a dedicated KV cache compression method. Extensive experiments on representative autoregressive models, MAGI-1 and SkyReels-V2, demonstrate the effectiveness of our approach. These results establish FlowCache as a new state-of-the-art for efficient, training-free acceleration, effectively bridging the gap between the theoretical promise of autoregressive video models and the practical feasibility of real-time, ultra-long video synthesis.

ACKNOWLEDGMENTS

This work is supported by the National Key Research and Development Program of China (No. 2025YFE0113500), the National Science Fund for Distinguished Young Scholars (No. 62025603), the Fundamental Research Funds for the Central Universities, the National Natural Sci- ence Foundation of China (No. 62576299, No. U21B2037, No. U22B2051, No. U23A20383, No. 62176222,

No. 62176223, No. 62176226, No. 62072386, No. 62072387, No. 62072389, No. 62002305, No. 62272401), and the Natural Science Foundation of Fujian Province of China (No. 2021J06003, No. 2022J06001).

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

## A  LLM Usage Statement

Large Language Models (LLMs) were used to aid in the writing and polishing of the manuscript. Specifically, we used an LLM to assist in refining the language, improving readability, and ensuring clarity in various sections of the paper. The model helped with tasks such as sentence rephrasing, grammar checking, and enhancing the overall flow of the text.

It is important to note that the LLM was not involved in the ideation, research methodology, or experimental design. All research concepts, ideas, and analyses were developed and conducted by the authors. The contributions of the LLM were solely focused on improving the linguistic quality of the paper, with no involvement in the scientific content or data analysis.

The authors take full responsibility for the content of the manuscript, including any text generated or polished by the LLM. We have ensured that the LLM-generated text adheres to ethical guidelines and does not contribute to plagiarism or scientific misconduct.

## B  Proof of Theorem 1

**Theorem 1** *Assume the diffusion model $v_\theta$ has converged to the optimal velocity field of the flow matching objective. Further, assume the scheduling function is of the power-law form $\sigma(t) = (t/T)^p$ for some constant power $p > 0$ and total time $T$.*

*Given $0 < t_1 < t_2 \leq T$ and a data chunk $X^i$ whose initial noise state $X_T^i$ is drawn from $\mathcal{N}(0, I)$, the following inequality holds:*

$$L1_{rel}(X, t_1, i) \geq L1_{rel}(X, t_2, i) \tag{13}$$

**Proof 1** *The relative L1 distance is defined as $L1_{rel}(X, t, i) = \frac{\|v_\theta(X_t^i, t, c) \cdot \Delta t\|_1}{\|X_t^i\|_1}$. To prove the theorem, we will analyze the monotonicity of this function with respect to time $t$.*

*First, according to the theorem's assumption that the model is optimal, we can replace $v_\theta$ with its ideal form in the flow matching framework:*

$$v_\theta(X_t^i, t, c) = -\frac{\sigma'(t)}{\sigma(t)}(X_t^i - X_0^i)$$

*Substituting this into the $L1_{rel}$ expression gives:*

$$L1_{rel}(X, t, i) = \frac{\left\| -\frac{\sigma'(t)}{\sigma(t)}(X_t^i - X_0^i) \cdot \Delta t \right\|_1}{\|X_t^i\|_1}$$

$$= \left| \frac{\sigma'(t)}{\sigma(t)} \right| \cdot \frac{\|X_t^i - X_0^i\|_1}{\|X_t^i\|_1} \cdot |\Delta t|$$

*Since $\sigma(t)$ is monotonically increasing for $t > 0$, both $\sigma'(t)$ and $\sigma(t)$ are positive, as is $\Delta t$. We can therefore remove the absolute value signs. Let's analyze the expression as a product of two time-dependent functions, $A(t)$ and $B(t)$:*

$$L1_{rel}(X, t, i) = \underbrace{\left( \frac{\sigma'(t)}{\sigma(t)} \right)}_{A(t)} \cdot \underbrace{\left( \frac{\|X_t^i - X_0^i\|_1}{\|X_t^i\|_1} \right)}_{B(t)} \cdot \Delta t$$

*Next, we analyze the behavior of $A(t)$ and $B(t)$ individually.*

***Analysis of*** $A(t)$***:*** *According to the theorem's assumption, $\sigma(t) = (t/T)^p$. Its derivative is $\sigma'(t) = \frac{p}{T}(\frac{t}{T})^{p-1} = \frac{p \cdot t^{p-1}}{T^p}$. Therefore, $A(t)$ becomes:*

$$A(t) = \frac{\sigma'(t)}{\sigma(t)} = \frac{p \cdot t^{p-1}/T^p}{t^p/T^p} = \frac{p}{t}$$

*For $t \in (0, T]$, the function $A(t) = p/t$ is a strictly monotonically decreasing function.*

**Analysis of** $B(t)$**:** *We expand the norms in $B(t)$ using the forward process definition $X_t^i = (1 - \sigma(t))X_0^i + \sigma(t)X_{noise}^i$. The numerator term becomes:*

$$\|X_t^i - X_0^i\|_1 = \|\sigma(t)(X_{noise}^i - X_0^i)\|_1 = \sigma(t)\|X_{noise}^i - X_0^i\|_1$$

*Since $\sigma(t)$ is monotonically increasing with $t$, the numerator is a monotonically increasing function of $t$.*

*The denominator is $\|X_t^i\|_1 = \|(1 - \sigma(t))X_0^i + \sigma(t)X_{noise}^i\|_1$. In high-dimensional latent spaces representing natural data (like video chunks), the structured data $X_0^i$ generally possesses a higher L1 norm than an unstructured Gaussian noise vector $X_{noise}^i$ of the same dimension. The interpolation $X_t^i$ represents a smooth transition from the data manifold to the noise distribution. Consequently, it is a standard observation that the L1 norm $\|X_t^i\|_1$ is a monotonically decreasing function as $t$ goes from $0$ to $T$.*

*Thus, $B(t)$ is the ratio of a monotonically increasing function (numerator) and a monotonically decreasing function (denominator). This implies that $B(t)$ itself is a monotonically increasing function of $t$.*

**Combined Analysis:** *We need to determine the behavior of the product $L1_{rel} \propto A(t) \cdot B(t)$. We have a strictly decreasing function $A(t) \sim 1/t$ and an increasing function $B(t)$. To show that their product is decreasing, we must argue that the decay of $A(t)$ dominates the growth of $B(t)$.*

*Let's consider the derivative of their product $f(t) = A(t)B(t)$:*

$$f'(t) = A'(t)B(t) + A(t)B'(t)$$

*Substituting $A(t) = p/t$ and $A'(t) = -p/t^2$, the condition becomes:*

$$f'(t) \leq 0 \quad \Longleftrightarrow \quad \frac{p}{t}B'(t) \leq \frac{p}{t^2}B(t) \quad \Longleftrightarrow \quad \frac{B'(t)}{B(t)} \leq \frac{1}{t}$$

*This inequality, stating that the relative growth rate of $B(t)$ is less than $1/t$, holds for a wide range of functions that exhibit sub-linear or saturating growth. Given that $B(t)$ starts from $B(0) = 0$ and increases towards a finite constant $B(T)$, its growth rate $B'(t)$ diminishes over time, while the term $1/t$ decays. The strong hyperbolic decay factor $A(t) = p/t$ from the scheduler's design effectively dominates the milder, saturating growth of the norm ratio $B(t)$. This ensures that their product is a monotonically decreasing function for $t \in (0, T]$.*

*Since $L1_{rel}(X, t, i)$ is proportional to this monotonically decreasing product, for any $0 < t_1 < t_2 \leq T$, we have:*

$$L1_{rel}(X, t_1, i) \geq L1_{rel}(X, t_2, i)$$

*This completes the proof.*

$\square$

## C    PROOF OF COROLLARY 1

**Corollary 1 (Cross-Chunk Divergence of Relative L1 Distance)** *Assuming that distinct video chunks $i \neq j$ represent heterogeneous content such that their state norms differ at any intermediate timestep $t \in (0, T)$ (i.e., $\|X_t^i\|_1 \neq \|X_t^j\|_1$), and that the model's update magnitude $\|v_\theta(X_t^k, t, c) \cdot \Delta t\|_1$ is approximately invariant across chunks for a fixed $t$, then their relative L1 distances are unequal:*

$$L1_{rel}(X, t, i) \neq L1_{rel}(X, t, j). \tag{14}$$

**Proof 2** *Our objective is to prove that $L1_{rel}(X, t, i) \neq L1_{rel}(X, t, j)$ for $i \neq j$ under the corollary's assumptions. We begin by recalling the definition from Equation 5 for two distinct chunks, $i$ and $j$:*

$$L1_{rel}(X, t, i) = \frac{\|v_\theta(X_t^i, t, c) \cdot \Delta t\|_1}{\|X_t^i\|_1}, \quad L1_{rel}(X, t, j) = \frac{\|v_\theta(X_t^j, t, c) \cdot \Delta t\|_1}{\|X_t^j\|_1}. \tag{15}$$

We analyze the ratio of these two quantities:

$$\frac{L1_{rel}(X,t,i)}{L1_{rel}(X,t,j)} = \frac{\|v_\theta(X_t^i,t,c) \cdot \Delta t\|_1}{\|v_\theta(X_t^j,t,c) \cdot \Delta t\|_1} \cdot \frac{\|X_t^j\|_1}{\|X_t^i\|_1}. \tag{16}$$

First, we consider the numerators of the original expressions. By the assumption of approximately constant update magnitude, the model's output norm is largely determined by the timestep $t$ rather than the specific chunk content. This is because conditioning information from prior chunks, processed through mechanisms like attention with Softmax normalization, primarily steers the direction of the update vector $v_\theta$ rather than its overall magnitude. Thus, we can posit that for a fixed $t$:

$$\|v_\theta(X_t^i,t,c) \cdot \Delta t\|_1 \approx \|v_\theta(X_t^j,t,c) \cdot \Delta t\|_1. \tag{17}$$

Therefore, the first term in Equation 16 is approximately equal to 1.

Next, we examine the denominators. The assumption of content heterogeneity implies that the target data distributions for each chunk, $X_0^i$ and $X_0^j$, are different. Since the flow matching process $x_t = (1 - \sigma(t))x_0 + \sigma(t)x_T$ defines a unique trajectory from the initial noise $x_T$ to the final data $x_0$, the distinctness of $X_0^i$ and $X_0^j$ guarantees that their entire denoising paths are different. Consequently, at any intermediate time $t \in (0, T)$, their states are unequal, leading to different L1 norms:

$$\|X_t^i\|_1 \neq \|X_t^j\|_1. \tag{18}$$

Substituting the approximation from Equation 17 into Equation 16, we get:

$$\frac{L1_{rel}(X,t,i)}{L1_{rel}(X,t,j)} \approx \frac{\|X_t^j\|_1}{\|X_t^i\|_1}. \tag{19}$$

Given the inequality in Equation 18, the right-hand side of this approximation cannot be equal to 1. It follows directly that $L1_{rel}(X,t,i) \neq L1_{rel}(X,t,j)$. This demonstrates that the relative L1 distance metric inherently diverges across chunks at a fixed timestep, reflecting their distinct underlying content and corresponding states along the denoising trajectory.

$\square$

## D  IMPLEMENTATION AND EFFICIENCY DETAILS

### D.1  IMPLEMENTATION DETAILS OF KV CACHE COMPRESSION

KV Cache compression begins only after the cache budget is reached. The process operates in two phases:

1. A **Cache Filling Phase:** New KV cache entries are added to the cache until the budget is reached, with no compression occurring during this phase.
2. A **Compression Phase:** Once the cache is full, the compression mechanism is triggered upon each new KV cache entry arrival. Our method then compresses existing entries within the budget to accommodate the new entry.

While KV cache compression is triggered upon reaching the specified budget, FlowCache also incorporates cache reuse for input video chunk activations, which applies to all video chunks throughout the generation pipeline. To assess the combined speedup, we measure end-to-end acceleration.

### D.2  EFFICIENCY ANALYSIS OF KV CACHE COMPRESSION AND FEATURE REUSE

The importance and similarity judgments during KV cache compression, as well as the input cache retention in FlowCache, may raise questions about increased memory usage during inference. However, our theoretical analysis and experimental results confirm that both types of additional overhead are negligible.

According to Equations 9 and 11 in the paper, KV cache compression introduces additional online dynamic overhead for computing KV cache importance. For Equation 9, we use only last 50 query

tokens to compute attention scores following previous work MInference (Jiang et al., 2024) and Flexprefill (Lai et al., 2025), allowing us to identify important elements in the KV cache without significant additional memory or computational overhead. For Equation 11, we derive an equivalent form that substantially reduces both additional memory (17.79 GB $\rightarrow$ 0.30 GB) and computational overhead (10.26s $\rightarrow$ 0.077s) by exchanging the order of matrix multiplication and summation:

$$\frac{1}{L_k}\sum_{i=1}^{L_k}S_{ij}^{(h)} = \underbrace{\frac{1}{L_k}\sum_{i=1}^{L_k}(K_{\text{clean}}^{(h)}K_{\text{clean}}^{(h)}{}^T)_{ij}}_{\text{Naive: 17.79 GB, 10.26s}} = \underbrace{\left(\frac{1}{L_k}\sum_{i=1}^{L_k}(K_{\text{clean}}^{(h)})_{ij}\right)K_{\text{clean}}^{(h)}{}^T}_{\text{Optimized: 0.30 GB, 0.077s}}, \quad (20)$$

where $K_{\text{clean}}^{(h)} \in \mathbb{R}^{L_k \times d}$. In the formula, memory and time costs are evaluated at $L_k = 24{,}300$ and $d = 128$ for illustrative purposes. Through this mathematical transformation, the memory and time overhead of our online dynamic KV cache importance computation becomes negligible.

For the input cache, given $X \in \mathbb{R}^{L_q \times \text{hidden\_states}}$, the caching cost at different timesteps is fixed. When $L_q = 108000$ and hidden_states $= 3072$, the corresponding memory footprint is only 0.6 GB, which is negligible in the overall inference overhead.

Regarding computational overhead, we profiled the FLOPs (floating-point operations) of the operation for selecting and retaining tokens in the KV cache. When generating a 10-second, 720×720 text-to-video sequence and with a KV cache budget of 5 chunks, this operation is performed only 5 times throughout the entire generation process. The actual computational overhead of this operation is 0.0013 PFLOPs with token-level query granularity and 0.05 PFLOPs with frame-level query granularity, respectively—both negligible compared to the total FLOPs of FlowCache-fast (140 PFLOPs).

The additional memory footprint and computational overhead introduced by FlowCache are therefore negligible. Memory usage experimental results are shown in Table 6 .

# E  ABLATION STUDY DETAILS

## E.1  APPLICATION ON THE 16-STEP DISTILLED MODEL.

Specifically, the VBench metrics we selected are: Imaging Quality, Aesthetic Quality, Motion Smoothness, Dynamic Degree, Background Consistency, Subject Consistency, Scene, Overall Consistency. Our evaluation consists of two parts: (1) benchmarking the 16-step distilled baseline against FlowCache-fast applied to the 64-step model, and (2) comparing TeaCache and FlowCache when applied to the 16-step distilled version. Key findings include:

**Distillation vs. Acceleration Trade-offs.** The 16-step distilled baseline achieves faster inference (2.38× vs. 3.56× speedup) but compromises quality compared to FlowCache-fast on the 64-step model (71.69% vs. 70.26%).

**Complementary Benefits.** FlowCache applied to the distilled model achieves superior speedup over TeaCache (1.92× vs. 1.17×) while preserving generation quality (70.72% vs. 70.60%).

These results in Table 3 demonstrate that FlowCache complements distillation techniques, offering orthogonal performance gains. Notably, FlowCache requires no large-scale distillation training of autoregressive models and operates as a plug-and-play online method.

## E.2  PERFORMANCE UNDER DIFFERENT KV CACHE COMPRESSION SETTINGS.

MAGI-1 (Teng et al., 2025) has conducted an investigation into the Key-Value (KV) range mechanism on the Physics-IQ Benchmark (Motamed et al., 2025). This task imposes stringent demands on both physical commonsense reasoning and temporal modeling capabilities. To produce physically plausible videos, the model must effectively leverage and retain sufficiently long historical context.

Motivated by this, our work systematically investigates how to maximally preserve critical historical context during KV cache compression in autoregressive video generation models, using the Physics-IQ benchmark as the evaluation setting.

First, we conduct a granular analysis of queries and keys during the KV cache compression process, specifically investigating: (1) the appropriate granularity at which queries should be used to select salient KV cache entries, and (2) the appropriate granularity for keys to be retrieved. For further details, please refer to E.2.1.

Second, We perform an ablation study on the hyperparameter $\lambda$ in Equation 12 to systematically examine how to balance importance and redundancy reduction in KV cache compression. For further details, please refer to E.2.2.

Third, we evaluate our method under varying KV cache budgets to quantitatively analyze the trade-off between video generation quality and GPU memory consumption. For further details, please refer to E.2.3.

### E.2.1 COMPRESSION GRANULARITY

Experimental results are summarized in Table 4. Under the baseline configuration without any acceleration or compression strategy, MAGI-1 achieves an accuracy of 47.60% on Physics-IQ. However, applying a non-discriminative global reuse strategy (TeaCache) causes a drastic performance drop to 13.69%, indicating that coarse-grained reuse severely degrades model performance. In contrast, the ChunkWise reuse strategy maintains a substantially higher accuracy of 43.10%, significantly outperforming TeaCache.

Our experiments further reveal that, when the query granularity is fixed at the token level (specifically, we select the last 50 tokens of the last denoising chunk as the query), the choice of key granularity has a pronounced impact on performance. Preserving keys at the token level—rather than aggregating them into frame- or chunk-level representations—enables finer-grained matching with historical context and yields higher accuracy (39.34% vs. 38.62% and 38.99%, respectively). This finding underscores the importance of maintaining fine-grained key representations during KV cache compression to retain historically relevant information critical for accurate prediction.

Moreover, when key granularity is fixed at the token level, using frame-level queries achieves higher accuracy (39.53%) compared to token-level queries (39.34%). This suggests that retrieving historical KV pairs at the semantically richer frame level facilitates more effective aggregation of context relevant to the current prediction, thereby better preserving physical consistency in the generated video. However, the experimental results indicate that this improvement is marginal, while using frame-level queries to retrieve important key-value (KV) cache entries incurs a substantial increase in GPU memory consumption. Therefore, we argue that using token-level queries to retrieve important KV cache entries is sufficient—a design choice that shares conceptual similarities with prior approaches such as Minference (Jiang et al., 2024) and FlexPrefill (Lai et al., 2025).

### E.2.2 HYPERPARAMETER $\lambda$

We conducted an ablation study on the hyperparameter $\lambda$ in Equation 12. Results in Table 5 demonstrate that Physics-IQ scores increase as $\lambda$ decreases, then stabilize beyond a threshold value. This trend indicates that redundancy between KV caches (Equation 11) more effectively captures useful information from past video frames than the raw importance metric (Equation 9) alone. Based on these findings, we adopt $\lambda = 0.07$ uniformly across all video types and experiments.

### E.2.3 DIFFERENT KV CACHE BUDGET

We have conducted additional experiments to explicitly demonstrate the memory-quality trade-offs. Notably, all experiments exclude cache reuse mechanisms for fair comparison. Specifically, we evaluated Physics-IQ scores across varying peak memory budgets during inference. The results in Table 6 show that FlowCache reduces peak memory by up to 33.6% while maintaining stable Physics-IQ scores (±0.1), even slightly outperforming Vanilla at 5 chunks budget. This demonstrates its effectiveness in achieving substantial memory savings without compromising physics reasoning quality. These findings provide concrete evidence of the trade-off between memory efficiency and generation quality, strengthening our contribution as claimed.

Table 3: Quantitative evaluation of inference efficiency and visual quality between 16-step and 64-step in MAGI-1. (VBench[*] refers to the weighted sum of the scores across the eight representative metrics we selected.)

| Model | Method | Steps | PFLOPs ↓ | Speedup ↑ | Latency(s) ↓ | VBench[*] ↑ |
|---|---|---|---|---|---|---|
| | Vanilla | 64 | 306 | 1× | 2873 | 69.12% |
| | FlowCache | 64 | 140 | 2.38× | 1209 | **71.69%** |
| MAGI-1 | Vanilla | 16 | 77 | 3.56× | 808 | 70.26% |
| | TeaCache | 16 | 74 | 4.16× | 691 | 70.60% |
| | FlowCache | 16 | 63 | 6.84× | 420 | **70.72%** |

Table 4: Ablation study of KV cache compression granularity for MAGI-1 on Physics-IQ Benchmark.

| Reuse Strategy | KV Cache Compression | Query Granularity | Key Granularity | Physics-IQ ↑ |
|---|---|---|---|---|
| - | - | - | - | 47.60% |
| TeaCache | - | - | - | 13.69% |
| ChunkWise | - | - | - | 43.10% |
| ChunkWise | Enabled | Token | Token | 39.34% |
| ChunkWise | Enabled | Token | Frame | 38.62% |
| ChunkWise | Enabled | Token | Chunk | 38.99% |
| ChunkWise | Enabled | Frame | Token | **39.53%** |

Table 5: Ablation study of hyperparameter $\lambda$ for MAGI-1 on Physics-IQ Benchmark.

| $\lambda$ | Physics-IQ ↑ |
|---|---|
| 0.03 | 39.38% |
| 0.07 | **39.53%** |
| 0.15 | 37.11% |
| 0.20 | 38.42% |

Table 6: Ablation study of different KV cache budget for MAGI-1 on Physics-IQ Benchmark.

| Method | KV Cache Budget (Chunks) | Peak Memory (GB) | Physics-IQ ↑ |
|---|---|---|---|
| MAGI-1 | 8 | 42.84 | 47.60% |
| FlowCache w/o cache reuse | 7 | 34.05 | 47.55% |
| FlowCache w/o cache reuse | 6 | 31.25 | 46.72% |
| FlowCache w/o cache reuse | 5 | 28.45 | 47.65% |

## F  VISUALIZATION

We visualize the qualitative results on both MAGI-1 and SkyReels-V2 in Figure 4 and Figure 5, reporting the specific speedup ratios and VBench scores for each method (Vanilla, TeaCache, FlowCache-slow, and FlowCache-fast). As observed, FlowCache outperforms TeaCache in terms of fine-grained details and overall image quality, avoiding the visible noise artifacts that TeaCache tends to introduce. Moreover, in videos featuring complex scenes or significant motion, both FlowCache variants demonstrate superior robustness compared to TeaCache. Furthermore, while FlowCache-

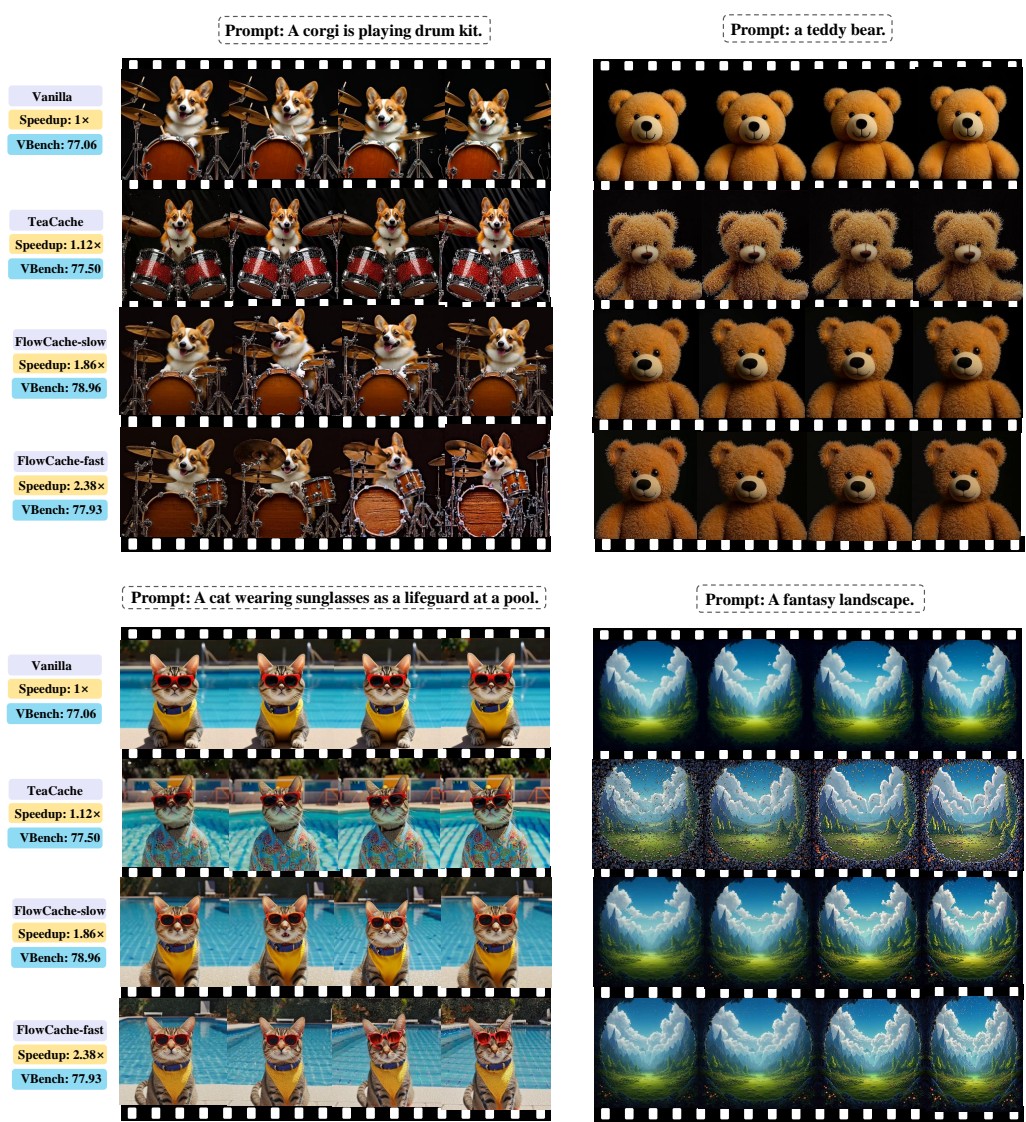

Figure 4: Qualitative results of text-to-video generation on MAGI-1. We present TeaCache, FlowCache-slow, FlowCache-fast, and the Vanilla model. The frames are randomly sampled from the generated video.

fast achieves higher acceleration, FlowCache-slow preserves higher fidelity, maintaining greater consistency with the original video generation.

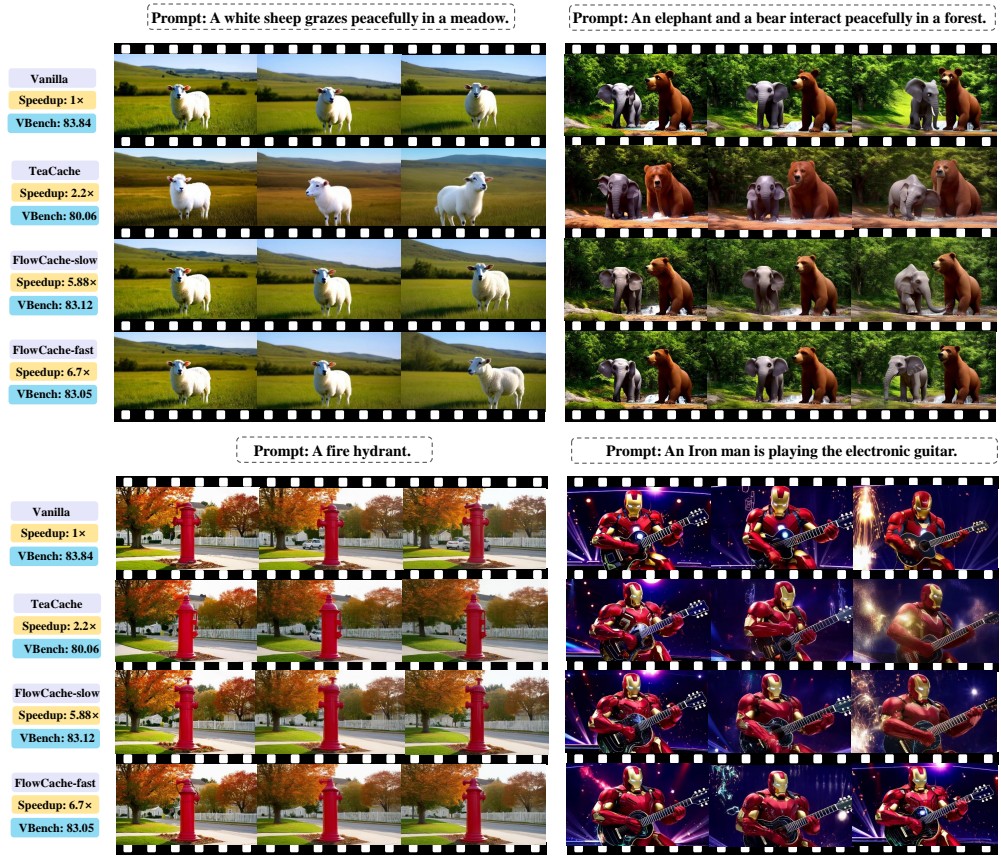

Figure 5: Qualitative results of text-to-video generation on SkyReels-V2. We present TeaCache, FlowCache-slow, FlowCache-fast, and the Vanilla model. The frames are randomly sampled from the generated video.

