# OpenReview forum: "Flow Caching for Autoregressive Video Generation"
_ICLR.cc/2026/Conference — ICLR 2026 Poster_

### Official Review · Reviewer_3Ldg · 2025-10-31

**Soundness:** 3
**Presentation:** 3
**Contribution:** 3
**Rating:** 6
**Confidence:** 4

**Summary:**

This paper proposes a **cache compression method** designed to significantly reduce the KV cache size in autoregressive (AR) video generation. The proposed approach effectively improves generation speed while largely maintaining output quality.

**Strengths:**

- The paper is **well-organized**, **logically structured**, and clearly highlights its main contributions.
- The proposed method introduces an **adaptive criterion** for KV cache compression, enabling dynamic adjustment of the compression ratio.
- This adaptive mechanism effectively improves cache compression efficiency and leads to faster video generation without significant quality degradation.

**Weaknesses:**

1. **Equation (8)** appears potentially ambiguous. It seems that the Softmax operation should be applied along the \( L_k \) dimension, but the current formulation places Softmax directly outside \( q_i \) and \( k_j \), which may be mathematically inconsistent.
2. It is unclear **when compression begins**—does it occur before reaching the cache budget, or only after exceeding it? If the method only keeps top-B tokens, then no compression happens before reaching the budget. The paper should clarify that the reported speedup comparisons are made **after the cache reaches the specified budget**.
3. Although the proposed approach achieves high compression rates, each compression step requires a **pre-computation of attention scores** to determine which tokens to keep, followed by another computation of the true attention scores for the retained tokens. This effectively adds an extra forward pass. The paper should report the **computational overhead ratio** introduced by this additional calculation.

**Questions:**

See weakness above

---

> ### Author Response · Authors · 2025-11-27
> **Response to Reviewer 3Ldg (Part: 1/2)**
>
> **W1: Equation (8) appears potentially ambiguous. It seems that the Softmax operation should be applied along the ( L_k ) dimension, but the current formulation places Softmax directly outside ( q_i ) and ( k_j ), which may be mathematically inconsistent.**
>
> **Reply:** We thank the reviewer for this keen observation. The reviewer is correct—the original formulation lacked precision and could lead to misinterpretation. The Softmax operation is indeed applied along the key dimension ($L_k$). We have corrected Equation (8) in the revised manuscript as follows:
>
> $$
> \\text{Imp} _ j^{(h)} = \\frac{1}{L _ q} \\sum _ {i=1}^{L _ q} \\left[ \\text{softmax} \\left( \\frac{q _ i^{(h)} (K _ {\\text{clean}}^{(h)})^\\top}{\\sqrt{d}} \\right) e _ j \\right].
> \\tag{1}
> $$
>
> Here, $\frac{q_i^{(h)} (K_{\text{clean}}^{(h)})^\top}{\sqrt{d}}$ computes attention scores between query $q_i^{(h)}$ and all keys in $K_{\text{clean}}^{(h)}$, with Softmax normalizing these scores over the $L_k$ dimension. The one-hot vector $e_j \in \mathbb{R}^{L_k \times 1}$ selects the $j$-th element from the normalized attention distribution, isolating the score for the $j$-th key. These scores are then averaged across all queries to obtain the final importance score. We have corrected this notation throughout the revised manuscript and expanded the explanation to clarify the derivation.
>
> **W2: It is unclear when compression begins—does it occur before reaching the cache budget, or only after exceeding it? If the method only keeps top-B tokens, then no compression happens before reaching the budget. The paper should clarify that the reported speedup comparisons are made after the cache reaches the specified budget.**
>
> **Reply:** We thank the reviewer for this insightful question, which highlights a critical aspect of our method's mechanics.
>
> To clarify, KV Cache compression begins only after the cache budget is reached. The process operates in two phases:
>
> 1. *Cache Filling Phase*: New KV cache entries are added to the cache until the budget is reached, with no compression occurring during this phase.
> 2. *Compression Phase*: Once the cache is full, the compression mechanism is triggered upon each new KV cache entry arrival. Our method then compresses existing entries within the budget to accommodate the new entry.
>
> While KV cache compression is triggered upon reaching the specified budget, FlowCache also incorporates cache reuse for input video chunk activations (see our response to Reviewer xZAs, W1), which applies to all video chunks throughout the generation pipeline. To assess the combined speedup, we measure end-to-end acceleration.
>
> We have clarified these points in the revised manuscript. We thank the reviewer for helping us improve the paper's clarity.

---

> ### Author Response · Authors · 2025-11-27
> **Response to Reviewer 3Ldg (Part: 2/2)**
>
> **W3: Although the proposed approach achieves high compression rates, each compression step requires a pre-computation of attention scores to determine which tokens to keep, followed by another computation of the true attention scores for the retained tokens. This effectively adds an extra forward pass. The paper should report the computational overhead ratio introduced by this additional calculation.**
>
> **Reply:** We thank the reviewer for this constructive feedback. To clarify, our attention score calculation is performed online after the KV cache exceeds the budget, without requiring an additional forward pass. We acknowledge that the importance and similarity judgments during KV cache compression, as well as the input cache retention in FlowCache, may raise questions about increased memory usage during inference. However, our theoretical analysis and experimental results confirm that both types of additional overhead are negligible.
>
> According to Equations 9 and 11 in the paper, KV cache compression introduces additional online dynamic overhead for computing KV cache importance. For Equation 9, we use only last 50 query tokens to compute attention scores following previous work MInference [1] and Flexprefill [2], allowing us to identify important elements in the KV cache without significant additional memory or computational overhead. For Equation 11, we derive an equivalent form that substantially reduces both additional memory (17.79 GB → 0.30 GB) and computational overhead (10.26s → 0.077s) by exchanging the order of matrix multiplication and summation:
>
> $$
> \\frac{1}{L _ k} \\sum _ {i=1}^{L _ k} S^{(h)} _ {ij} =
> \\underbrace{\\frac{1}{L _ k} \\sum _ {i=1}^{L _ k} (K _ {\\text{clean}}^{(h)} {K _ {\\text{clean}}^{(h)}}^T) _ {ij}} _ {\\text{Naive: 17.79 GB, 10.26s}}
> = \\underbrace{\\left(\\frac{1}{L _ k} \\sum _ {i=1}^{L _ k} (K _ {\\text{clean}}^{(h)}) _ {ij}\\right) {K _ {\\text{clean}}^{(h)}}^T} _ {\\text{Optimized: 0.30 GB, 0.077s}},
> \\tag{2}
> $$
>
> where $K_{\text{clean}}^{(h)} \in \mathbb{R}^{L_k \times d}$. In the formula, memory and time costs are evaluated at $L_k=24{,}300$ and $d=128$ for illustrative purposes. Through this mathematical transformation, the memory and time overhead of our online dynamic KV cache importance computation becomes negligible.
>
> For the input cache, given $X \in \mathbb{R}^{L_q \times \text{hidden\\_states}}$, the caching cost at different timesteps is fixed. When $L_q = 108000$ and $\text{hidden\\_states}=3072$, the corresponding memory footprint is only 0.6 GB, which is negligible in the overall inference overhead.
>
> Regarding computational overhead, we profiled the FLOPs (floating-point operations) of the operation for selecting and retaining tokens in the KV cache. When generating a 10-second, 720×720 text-to-video sequence and with a KV cache budget of 5 chunks, this operation is performed only 5 times throughout the entire generation process. The actual FLOPs overhead of this operation with two different query granularity is as follows, while the total FLOPs of FlowCache-Fast is 140 P, making this overhead practically negligible.
>
> | Query granularity | FLOPS (P)      |
> |-------------------|----------------|
> | Token            | 0.0013         |
> | Frame             | 0.05           |
>
>
> The additional memory footprint and computational overhead introduced by FlowCache are therefore negligible. Memory usage experimental results are shown in the following table:
>
> | Method                            | KV Cache Budget (Chunks) | Peak Memory (GB) |
> |-----------------------------------|--------------------------|------------------|
> | MAGI-1                 | 8                        | 42.84            |
> | FlowCache                         | 7                        | 34.05            |
> | FlowCache                         | 6                        | 31.25            |
> | FlowCache              | 5                        | 28.45            |
>
>
> **References**
>
> [1] Jiang et al., Minference 1.0: Accelerating pre-filling for long-context llms via dynamic sparse attention. NeurIPS 2024.
>
> [2] Lai et al., Flexprefill: A context-aware sparse attention mechanism for efficient long-sequence inference. ICLR 2025.

---

### Official Review · Reviewer_fkWp · 2025-11-01

**Soundness:** 3
**Presentation:** 3
**Contribution:** 4
**Rating:** 8
**Confidence:** 4

**Summary:**

This paper introduces FlowCache, a novel caching and compression framework for autoregressive video generation that addresses the unique challenge of heterogeneous chunk denoising states. The method is theoretically grounded, empirically strong, and significantly advances the state of the art in efficient long-video synthesis.

**Strengths:**

1.  The paper's primary strength is its clear identification and empirical validation of *why* existing caching methods fail for autoregressive models: the "heterogeneous denoising states." This is a sharp and important insight.
2. The "chunkwise caching" policy is a direct, logical, and highly effective solution to the problem identified. The ablation study decisively proves that this chunk-specific strategy is the main reason for the method's success in preserving quality.
3. Additionally, the KV cache compression method is also well-motivated. It correctly identifies the high-redundancy problem in video data and proposes a solution that intelligently balances both importance and redundancy, which is a clear improvement over importance-only methods from language modeling. From my perspective, this is an excellent work overall.

**Weaknesses:**

1. There is an internal contradiction between the paper's theory and its empirical results. **Theorem 1** (and its proof in Appendix B) is used to establish that the relative L1 distance is a *monotonically decreasing* function of time (as $t$ goes from 0 to T). However, the paper's *own empirical plot* (Figure 2) and *main text* (e.g., "relative L1 distance monotonically increases as denoising progresses" in line 302) show the exact opposite. This contradiction undermines the stated theoretical foundation.
2. The ablation in Table 2 suggests that the complex KV cache compression adds very little performance on top of the main chunkwise policy. On MAGI-1, the chunkwise policy *alone* achieves a 77.66% VBench score, while the *full* method with compression gets 77.93%. This small benefit may not justify the added complexity of the importance-redundancy scoring mechanism; more discussion is required.

**Questions:**

How sensitive is the KV cache compression performance to the choice of λ and the granularity settings? Is there a recommended configuration for different video types?

---

> ### Author Response · Authors · 2025-11-27
> **Response to Reviewer fkWp (Part: 1/2)**
>
> **W1: There is an internal contradiction between the paper's theory and its empirical results. Theorem 1 (and its proof in Appendix B) is used to establish that the relative L1 distance is a monotonically decreasing function of time (as t goes from 0 to T). However, the paper's own empirical plot (Figure 2) and main text (e.g., "relative L1 distance monotonically increases as denoising progresses" in line 302) show the exact opposite. This contradiction undermines the stated theoretical foundation.**
>
> **Reply:**   We thank the reviewer for this careful observation and welcome the opportunity to clarify this misunderstanding.
>
> Figure 2 describe the reverse denoising process, where t decreases from T to 0 as denoising progresses, instead of increasing from 0 to T. Thus, our empirical observations fully align with Theorem 1.
>
> **W2: The ablation in Table 2 suggests that the complex KV cache compression adds very little performance on top of the main chunkwise policy. On MAGI-1, the chunkwise policy alone achieves a 77.66% VBench score, while the full method with compression gets 77.93%. This small benefit may not justify the added complexity of the importance-redundancy scoring mechanism; more discussion is required.**
>
> **Reply:** The primary contribution of FlowCache is to achieve significant speedup while $\textbf{preserving}$ the quality of generated videos. As a training-free acceleration method, our work focuses on inference efficiency rather than enhancing video generation quality. Moreover, the importance and similarity judgments during KV cache compression, as well as the input cache retention in FlowCache, may raise questions about increased memory usage during inference. However, our theoretical analysis and experimental results confirm that both types of additional overhead are negligible.
>
> According to Equations 9 and 11 in the paper, KV cache compression introduces additional online dynamic overhead for computing KV cache importance. For Equation 9, we use only last 50 query tokens to compute attention scores following previous work MInference [1] and Flexprefill [2], allowing us to identify important elements in the KV cache without significant additional memory or computational overhead. For Equation 11, we derive an equivalent form that substantially reduces both additional memory (17.79 GB → 0.30 GB) and computational overhead (10.26s → 0.077s) by exchanging the order of matrix multiplication and summation:
>
> $$
> \\frac{1}{L _ k} \\sum _ {i=1}^{L _ k} S^{(h)} _ {ij} =
> \\underbrace{\\frac{1}{L _ k} \\sum _ {i=1}^{L _ k} (K _ {\\text{clean}}^{(h)} {K _ {\\text{clean}}^{(h)}}^T) _ {ij}} _ {\\text{Naive: 17.79 GB, 10.26s}}
> = \\underbrace{\\left(\\frac{1}{L _ k} \\sum _ {i=1}^{L _ k} (K _ {\\text{clean}}^{(h)}) _ {ij}\\right) {K _ {\\text{clean}}^{(h)}}^T} _ {\\text{Optimized: 0.30 GB, 0.077s}},
> \\tag{1}
> $$
>
> where $K_{\text{clean}}^{(h)} \in \mathbb{R}^{L_k \times d}$. In the formula, memory and time costs are evaluated at $L_k=24{,}300$ and $d=128$ for illustrative purposes. Through this mathematical transformation, the memory and time overhead of our online dynamic KV cache importance computation becomes negligible.
>
> For the input cache, given $X \in \mathbb{R}^{L_q \times \text{hidden\\_states}}$, the caching cost at different timesteps is fixed. When $L_q = 108000$ and $\text{hidden\\_states}=3072$, the corresponding memory footprint is only 0.6 GB, which is negligible in the overall inference overhead.
>
> The additional memory footprint and computational overhead introduced by FlowCache are therefore negligible. Memory usage experimental results are shown in the following table:
>
> | Method                            | KV Cache Budget (Chunks) | Peak Memory (GB) |
> |-----------------------------------|--------------------------|------------------|
> | MAGI-1                 | 8                        | 42.84            |
> | FlowCache                         | 7                        | 34.05            |
> | FlowCache                         | 6                        | 31.25            |
> | FlowCache              | 5                        | 28.45            |

---

> ### Author Response · Authors · 2025-11-27
> **Response to Reviewer fkWp (Part: 2/2)**
>
> **Q1: How sensitive is the KV cache compression performance to the choice of λ and the granularity settings? Is there a recommended configuration for different video types?**
>
> **Reply:** We thank the reviewer for this constructive question. We conducted an ablation study on the hyperparameter $\lambda$ in Equation 12. Results demonstrate that Physics-IQ scores increase as $\\lambda$ decreases, then stabilize beyond a threshold value. This trend indicates that redundancy between KV caches (Equation 11) more effectively captures useful information from past video frames than the raw importance metric (Equation 9) alone. Based on these findings, we adopt $\lambda = 0.07$ uniformly across all video types and experiments. Detailed ablation results are presented in the table below:
>
> | lambda λ | Physics-IQ |
> | :--- | :--- |
> | 0.03 | 39.38% |
> | 0.07 | 39.53% |
> | 0.15 | 37.11% |
> | 0.2 | 38.42% |
>
> Furthermore, in the appendix of our original paper, we conducted experiments evaluating different granularities for queries and keys. Specifically, our findings are as follows:
> 1. **Key Granularity:**  Token-level keys significantly outperform coarser (frame/chunk-level) keys (39.34% vs. 38.62% and 38.99%, respectively), highlighting the need to preserve fine-grained key information during KV cache compression.
>
> 2. **Query Granularity:** Though frame-level queries yield a marginal accuracy gain (39.53% vs. 39.34%), they incur high memory overhead. Thus, token-level queries are preferred—striking a better trade-off between performance and efficiency, and aligning with prior work (e.g., Minference[1], FlexPrefill[2]).
>
> Detailed ablation results are presented in the table below:
>
> | Query Granularity | Key Granularity | Physics-IQ |
> | :--- | :--- | :--- |
> | Token | Token | 39.34% |
> | Token | Frame | 38.62% |
> | Token | Chunk | 38.99% |
> | Frame | Token | 39.53% |
>
> Finally, for a recommended configuration for different video types, we find that, in KV cache compression for video generation, adopting λ = 0.07 along with token-level queries and token-level keys can achieve relatively strong performance.
>
> **References**
>
> [1] Jiang et al., Minference 1.0: Accelerating pre-filling for long-context llms via dynamic sparse attention. NeurIPS 2024.
>
> [2] Lai et al., Flexprefill: A context-aware sparse attention mechanism for efficient long-sequence inference. ICLR 2025.

---

### Official Review · Reviewer_LS46 · 2025-11-01

**Soundness:** 2
**Presentation:** 1
**Contribution:** 2
**Rating:** 2
**Confidence:** 5

**Summary:**

The paper studies training-free acceleration for autoregressive video diffusion. The authors empirically demonstrate that different chunks should have independent feature caching policies rather than a single global caching policy. A redundancy-aware KV-cache compression scheme is also adopted for long-video generation.
Experiments on MAGI-1 and SkyReels-V2 demonstrate a noticeable speedup with a minor drop in VBench score.

**Strengths:**

- The paper proposed training-free, plug-in acceleration for causal video diffusion. Treating each video chunk as its own, with an independent reuse policy, is well motivated by the observed heterogeneity across chunks at the same timesteps.
- Experiment isolates the benefit of chunkwise feature reuse over full reuse and shows that kv-compression has a small impact.

**Weaknesses:**

- Memory claims lack evidence. Memory usage is stated to be fixed, but no benchmark on memory usage is exhibited in the paper.
- Lack of quality comparison. Only a few images are displayed in the paper. No video clip was provided, making it hard to evaluate the visual quality.
- Lack of evaluation on long-video benchmark, e.g., VBench-long, since the method is claimed to be helpful for long-video generation.
- MAGI-1 already applied window attention (8-second preceding video content). Weakening the motivation for KV-cache compression.
- MAGI-1 has a shortcut step-distill version. The proposed method does not compare with it, nor apply the feature reuse to it.
- Feature reuse necessarily increases memory consumption because the cache has to be retained. This seems to conflict with the stated motivation of KVcache compression, which is to reduce memory usage.

**Questions:**

- line 99-101: Please clarify the experimental/testing setup for this result (e.g., GPU type, memory, batch size, video length). Without this, it’s hard to judge how generalizable the observation is.
- The paper claims to “provide insights into memory–quality trade-offs,” but the experiments do not actually show memory vs. quality curves/tables (e.g., VBench vs. cache size/compression ratio). This weakens the contribution.
- How is the chunkwise caching policy obtained in practice? Is it derived offline from a calibration set, or learned/heuristic? When generating videos of different lengths or motion patterns, does the policy need to be recomputed or adapted?
- Do you have numerical benchmarks (peak memory, KV size per frame/chunk, vs. baseline) to substantiate the claim of reducing the memory with KV-cache compression?
- misc:
  - The citation of [1] is not about diffusion and seems unrelated to the context in which it is cited.
  - VBench is a scaled score, not a percentage

[1] Mengwei Xu, et la, Deepcache: Principled cache for mobile deep vision. 2018

---

> ### Author Response · Authors · 2025-11-27
> **Response to Reviewer LS46 (Part: 1/4)**
>
> **W1: Memory claims lack evidence. Memory usage is stated to be fixed, but no benchmark on memory usage is exhibited in the paper.**
>
> **Reply:** We thank the reviewer for raising this concern. We acknowledge that the importance and similarity judgments during KV cache compression, as well as the input cache retention in FlowCache, may raise questions about increased memory usage during inference. However, our theoretical analysis and experimental results confirm that both types of additional overhead are negligible.
>
> According to Equations 9 and 11 in the paper, KV cache compression introduces additional online dynamic overhead for computing KV cache importance. For Equation 9, we use only last 50 query tokens to compute attention scores following previous work MInference [1] and Flexprefill [2], allowing us to identify important elements in the KV cache without significant additional memory or computational overhead. For Equation 11, we derive an equivalent form that substantially reduces both additional memory (17.79 GB → 0.30 GB) and computational overhead (10.26s → 0.077s) by exchanging the order of matrix multiplication and summation:
>
> $$
> \\frac{1}{L _ k} \\sum _ {i=1}^{L _ k} S^{(h)} _ {ij} =
> \\underbrace{\\frac{1}{L _ k} \\sum _ {i=1}^{L _ k} (K _ {\\text{clean}}^{(h)} {K _ {\\text{clean}}^{(h)}}^T) _ {ij}} _ {\\text{Naive: 17.79 GB, 10.26s}}
> = \\underbrace{\\left(\\frac{1}{L _ k} \\sum _ {i=1}^{L _ k} (K _ {\\text{clean}}^{(h)}) _ {ij}\\right) {K _ {\\text{clean}}^{(h)}}^T} _ {\\text{Optimized: 0.30 GB, 0.077s}},
> \\tag{1}
> $$
>
> where $K_{\text{clean}}^{(h)} \in \mathbb{R}^{L_k \times d}$. In the formula, memory and time costs are evaluated at $L_k=24{,}300$ and $d=128$ for illustrative purposes. Through this mathematical transformation, the memory and time overhead of our online dynamic KV cache importance computation becomes negligible.
>
> For the input cache, given $X \\in \\mathbb{R}^{L_q \\times \\text{hidden\\_states}}$, the caching cost at different timesteps is fixed. When $L_q = 108000$ and $\\text{hidden\\_states}=3072$, the corresponding memory footprint is only 0.6 GB, which is negligible in the overall inference overhead.
>
> The additional memory footprint and computational overhead introduced by FlowCache are therefore negligible. Memory usage experimental results are shown in the following table:
>
> | Method                            | KV Cache Budget (Chunks) | Peak Memory (GB) |
> |-----------------------------------|--------------------------|------------------|
> | MAGI-1                 | 8                        | 42.84            |
> | FlowCache                         | 7                        | 34.05            |
> | FlowCache                         | 6                        | 31.25            |
> | FlowCache       | 5                        | 28.45            |
>
> **W2: Lack of quality comparison. Only a few images are displayed in the paper. No video clip was provided, making it hard to evaluate the visual quality.**
>
> **Reply:** Thank you for your suggestions! In addition to the video clips on the first page, we provide supplementary video clips in the Appendix. All clips are also accessible via the following anonymous GitHub link: https://anonymous.4open.science/r/FlowCache-23495iclrAnonymous/Visualization/VIDEO_COMPARISON.md. These additional visualization results further demonstrate that FlowCache maintains superior video quality while achieving higher speedup ratios.
>
> **W3: Lack of evaluation on long-video benchmark, e.g., VBench-long, since the method is claimed to be helpful for long-video generation.**
>
> **Reply:** We thank the reviewer for raising this important point. We clarify that our evaluation was indeed conducted on the VBench-long benchmark to validate our method's effectiveness for long-video generation. To avoid any confusion, we have added an explicit note in the ''Evaluation Metrics'' of experimental settings section specifying that all VBench evaluations reported in this paper were performed using the VBench-long benchmark.

---

> ### Author Response · Authors · 2025-11-27
> **Response to Reviewer LS46 (Part: 2/4)**
>
> **W4: MAGI-1 already applied window attention (8-second preceding video content). Weakening the motivation for KV-cache compression.**
>
> **Reply:** We appreciate this question. While MAGI-1 employs window attention with adjustable window size and historical context length, prior work has established that extended historical information is essential for autoregressive video models to generate physically plausible videos. Figure 18 of MAGI-1 demonstrates this empirically: longer KV ranges yield higher Physics-IQ scores [4], a metric assessing conformance to physical laws across mechanics, optics, thermodynamics, and magnetism. The fundamental constraint is memory: extended KV caches incur substantial overhead due to long token sequences.
>
> FlowCache addresses this trade-off through selective compression of global historical information. Under identical memory budgets, our approach outperforms MAGI-1's window attention KV cache strategy, which retains only the noise-proximal KV cache for subsequent attention computations (47.65 vs. 45.50). Notably, all experiments exclude cache reuse mechanisms for fair comparison, as detailed in the table below:
>
> |   Method   |  KV Range   | Physics-IQ  |
> | ---------- | ----------- | ----------- |
> |   MAGI-1   |    8        | 47.60       |
> |  MAGI-1    |    5        | 45.50       |
> | FlowCache  |    5        | 47.65       |
>
> **W5: MAGI-1 has a shortcut step-distill version. The proposed method does not compare with it, nor apply the feature reuse to it.**
>
> **Reply:** We appreciate this suggestion. Following your recommendation, we conducted comprehensive experiments on the 16-timestep distilled MAGI-1 model. Due to computational constraints, we evaluated performance using representative VBench metrics selected based on established practices in video generation compression research [5,6,7,8]. To facilitate an intuitive comparison, we compute the average scores of the selected VBench metrics using the official normalization and weighting methodology provided by the VBench benchmark.
>
> Our evaluation consists of two parts: (1) benchmarking the 16-step distilled baseline against FlowCache-fast applied to the 64-step model, and (2) comparing TeaCache and FlowCache when applied to the 16-step distilled version. Key findings include:
>
> 1. **Distillation vs. Acceleration Trade-offs:** The 16-step distilled baseline achieves faster inference (2.38× vs. 3.56× speedup) but compromises quality compared to FlowCache-fast on the 64-step model (71.69% vs. 70.26%).
>
> 2. **Complementary Benefits:** FlowCache applied to the distilled model achieves superior speedup over TeaCache (6.84× vs. 4.16×) while preserving generation quality (70.72% vs. 70.60%).
>
> These results demonstrate that FlowCache complements distillation techniques, offering orthogonal performance gains. Notably, FlowCache requires no large-scale distillation training of autoregressive models and operates as a plug-and-play online method. Detailed results are presented in the table below:
>
> | Method | Steps | FLOPS(P) | Latency (↓) | Speedup (↑) | Imaging Quality | Aesthetic Quality | Motion Smoothness | Dynamic Degree | Background Consistency | Subject Consistency | Scene | Overall Consistency | Avg |
> | :--- | :--- | :--- | :--- | :--- | :--- | :--- | :--- | :--- | :--- | :--- | :--- | :--- | :--- |
> | Vanilla | 64 | 306 | 2873 | 1× | 60.77% | 60.25% | 99.58% | 12.78% | 98.60% | 98.46% | 21.25% | 25.28% | 69.12% |
> | FlowCache-fast | 64 | 140 | 1209 | ~2.38× | 63.69% | 62.40% | 98.15% | 38.89% | 97.96% | 96.38% | 25.86% | 26.63% | **71.69%** |
> | Vanilla | 16 | 77 | 808 | ~3.56× | 61.03% | 61.59% | 99.41% | 24.44% | 98.66% | 98.30% | 21.56% | 25.79% | 70.26% |
> | TeaCache | 16 | 74 | 691 | ~4.16× | 64.06% | 62.40% | 99.19% | 18.06% | 98.04% | 97.90% | 24.06% | 26.13% | 70.60% |
> | Flowcache | 16 | 63 | 420 | ~6.84× | 65.27% | 62.91% | 98.77% | 20.28% | 97.94% | 97.87% | 22.83% | 26.55% | **70.72%** |
>
> **W6: Feature reuse necessarily increases memory consumption because the cache has to be retained. This seems to conflict with the stated motivation of KVcache compression, which is to reduce memory usage.**
>
> **Reply:** As discussed in W1, the memory footprint of the input cache remains constant across different timesteps. Given $X \\in \\mathbb{R}^{L_q \\times \\text{hidden\\_states}}$, the caching cost at different timesteps is fixed. When $L_q = 108000$ and $\\text{hidden\\_states}=3072$, the corresponding memory footprint is only 0.6 GB, which is negligible in the overall inference overhead.

---

> ### Author Response · Authors · 2025-11-27
> **Response to Reviewer LS46 (Part: 3/4)**
>
> **Q1: line 99-101: Please clarify the experimental/testing setup for this result (e.g., GPU type, memory, batch size, video length). Without this, it’s hard to judge how generalizable the observation is.**
>
> **Reply:** We appreciate this suggestion and have revised lines 99-101 to include the requested experimental details:
>
> "For example, using an A800 GPU with batch size 1, generating a 10-second video at 720×720 resolution with full KV cache reference requires approximately 32GB of memory and 50 minutes of inference time for the MAGI-1 model."
>
> **Q2: The paper claims to ''provide insights into memory–quality trade-offs,'' but the experiments do not actually show memory vs. quality curves/tables (e.g., VBench vs. cache size/compression ratio). This weakens the contribution.**
>
> **Reply:** Thank you for this valuable suggestion. We have conducted additional experiments to explicitly demonstrate the memory-quality trade-offs. Specifically, we evaluated Physics-IQ scores across varying peak memory budgets during inference. The results show that FlowCache reduces peak memory by up to 33.6% while maintaining stable Physics-IQ scores (±0.1%), even slightly outperforming Vanilla at 5 chunks budget. This demonstrates its effectiveness in achieving substantial memory savings without compromising physics reasoning quality. These findings provide concrete evidence of the trade-off between memory efficiency and generation quality, strengthening our contribution as claimed. The detailed results are presented in the table below:
>
> | Method     | KV Cache Budget (Chunks)   | Peak memory (GB) | Physics-IQ |
> |------------|------------------------|------------------|---------|
> | Vanilla    | 8                | 42.84            | 47.60%   |
> | FlowCache  | 7                | 34.05            | 47.55%   |
> | FlowCache  | 6                | 31.25            | 46.72%   |
> | FlowCache  | 5                | 28.45            | 47.65%   |
>
>
> **Q3: How is the chunkwise caching policy obtained in practice? Is it derived offline from a calibration set, or learned/heuristic? When generating videos of different lengths or motion patterns, does the policy need to be recomputed or adapted?**
>
> **Reply:** Thank you for this careful observation! We clarify the missing implementation details here. Following Teacache, we employ a threshold-based criterion to determine cache reuse at each timestep. Note that, the L1 similarity metric is accumulated independently within each video chunk in FlowCache. Specifically, for the $i$-th video chunk:
>
> $$
> \\mathcal{f}(X, t, i) =
> \\begin{cases}
>     0 & \\text{if } t \\in (T-m, T], \\\\
>     0 & \\text{if } t \\in [0, T-m] \\text{ and } \\mathcal{f}(X, t+1, i) + L1_{\\mathrm{rel}}(X, t, i) > \\epsilon, \\\\
>     \\mathcal{f}(X, t+1, i) + L1_{\\mathrm{rel}}(X, t, i) & \\text{if } t \\in [0, T-m] \\text{ and } \\mathcal{f}(X, t+1, i) + L1_{\\mathrm{rel}}(X, t, i) \\leq \\epsilon,
> \\end{cases}
> \\tag{2}
> $$
>
> where $\epsilon$ denotes the threshold value and $m$ represents the number of initial timesteps excluded from cache reuse ($m=5$ for MAGI-1, $m=4$ for SkyReels-V2). When $\mathcal{f}(X, t, i) = 0$, forward computation is performed for $i$-th video chunk; otherwise, cached activations are reused. Our empirical analysis demonstrates that excluding early timesteps is critical for preserving generation quality. The threshold values are: MAGI-slow: 0.01, MAGI-fast: 0.015, SkyReels-V2-slow: 0.1, SkyReels-V2-fast: 0.15. We have incorporated the above description into the end of section 3.2 of the paper to help readers understand the details of FlowCache's solution.
>
> To ensure fair comparison across methods under comparable FLOPs, we profile the FLOPs of the autoregressive video model and establish a threshold $\epsilon$ accordingly. This process requires neither offline inference on a calibration set nor any learnable components. Notably, the thresholds $\epsilon$ and $m$ remain fixed across both our fast and slow versions, regardless of video length or motion complexity.

---

> ### Author Response · Authors · 2025-11-27
> **Response to Reviewer LS46 (Part: 4/4)**
>
> **Q4: Do you have numerical benchmarks (peak memory, KV size per frame/chunk, vs. baseline) to substantiate the claim of reducing the memory with KV-cache compression?**
>
> **Reply:** Please refer to our response to W1, W4 and Q2 for detailed numerical benchmarks comparing memory usage and quality metrics against baselines. We have included tables in the Appendix of the paper to substantiate our claims regarding memory reduction through KV-cache compression.
>
> **Q5: The citation of [3] is not about diffusion and seems unrelated to the context in which it is cited. VBench is a scaled score, not a percentage.**
>
> **Reply:**
>
> **Citation Error:** We thank the reviewer for identifying this error. The citation [3] incorrectly referenced an unrelated 2018 paper. The intended reference is Ma et al., "DeepCache: Accelerating Diffusion Models for Free" (2023), which directly addresses acceleration techniques for diffusion models. This has been corrected in the revised manuscript.
>
> **VBench Score Representation:** The reviewer correctly notes that VBench produces scaled scores rather than inherent percentages. We appreciate the opportunity to clarify our reporting methodology.
>
> VBench's evaluation pipeline computes normalized dimension scores (scaled 0–1), aggregates them into Quality and Semantic Scores, and produces a Total Score as their weighted average, typically ranging from 0 to 1. We present these scores as percentages for clarity and consistency with established reporting conventions in recent literature. Specifically, TeaCache [9] (Table 1), Toca [10] (Table 2), and AdaCache [11] (Table 1) all report VBench results as percentages. Our presentation aligns with these community practices while remaining faithful to the underlying normalized scores.
>
> **References**
>
> [1] Jiang et al., Minference 1.0: Accelerating pre-filling for long-context llms via dynamic sparse attention. NeurIPS 2024.
>
> [2] Lai et al., Flexprefill: A context-aware sparse attention mechanism for efficient long-sequence inference. ICLR 2025.
>
> [3] Xu et al., Deepcache: Principled cache for mobile deep vision. 2018
>
> [4] Motamed et al., Do generative video models understand physical principles? 2025
>
> [5] Zhao et al., ViDiT-Q: Efficient and Accurate Quantization of Diffusion Transformers for Image and Video Generation. ICLR 2025.
>
> [6] Feng et al., Q-VDiT: Towards Accurate Quantization and Distillation of Video-Generation Diffusion Transformers. ICML 2025.
>
> [7] Xu et al., VETA-DiT: Variance-Equalized and Temporally Adaptive Quantization for Efficient 4-bit Diffusion Transformers. NeurIPS 2025.
>
> [8] Shao et al., TR-DQ: Time-Rotation Diffusion Quantization. 2025.
>
> [9] Liu et al., Timestep Embedding Tells: It's Time to Cache for Video Diffusion Model. CVPR 2025.
>
> [10] Zou et al., ToCa: Accelerating Diffusion Transformers with Token-wise Feature Caching. ICLR 2025.
>
> [11] Kahatapitiya et al., Adaptive Caching for Faster Video DiTs. ICCV 2025.

---

### Official Review · Reviewer_xZAs · 2025-11-01

**Soundness:** 3
**Presentation:** 3
**Contribution:** 3
**Rating:** 6
**Confidence:** 4

**Summary:**

This paper proposed a caching framework for autoregressive video generation. The key idea is to change existing uniform caching strategies used in common video diffusion models. Firstly, it applies dynamic caching strategies to different noise levels. Higher noise level is more likely to reuse cached feature, while lower noise level is more likely to recompute. It also introduce a compression mechanism to kv cache due to large memory consumption based on importance and redundancy.

**Strengths:**

- The dynamic assignment of calculate or reuse significant improves the efficiency of the model i.e., more than 2x comparing to TeaCache-fast.
- The proposed KV cache compression method balances both past token visual importance and redundancy. It is specifically designed for video token characteristics.

**Weaknesses:**

- The idea of reuse or recompute based on L1 similarity  of L1rel is clearly stated and proved, but the detail of how to decide to reuse or recompute the cache is not clear. Is there any threshold or decision making module for this part?
- It is not clear why the proposed method both out perform the baseline method, TeaCache both in terms of speed and video quality. The reuse operation accelerates the speed, but why it also achieve better frame quality. It would better to have more detailed comparison and discussion between the baseline method.

**Questions:**

- For the dynamic chunk caching and reuse mechanism, is it adaptively applied across different videos based on the degree of motion dynamics? As mentioned in Weakness 1, the decision-making process remains unclear — it would be helpful to provide a concrete inference example illustrating how this adaptation works.
- Are there any known failure cases for the two proposed designs? For instance, how do they perform on videos with large or complex motion? In such cases, is the speed up improvement less significant?

---

> ### Author Response · Authors · 2025-11-27
> **Response to Reviewer xZAs (Part: 1/2)**
>
> **W1: The detail of how to decide to reuse or recompute the cache is not clear. Is there any threshold or decision making module for this part?**
>
> **Reply:** Thank you for this observation. We clarify the missing implementation details here. Following Teacache, we employ a threshold-based criterion to determine cache reuse at each timestep. Note that, the L1 similarity metric is accumulated independently within each video chunk in FlowCache. Specifically, for the $i$-th video chunk:
>
> $$
> \\mathcal{f}(X, t, i) =
> \\begin{cases}
>     0 & \\text{if } t \\in (T-m, T], \\\\
>     0 & \\text{if } t \\in [0, T-m] \\text{ and } \\mathcal{f}(X, t+1, i) + L1_{\\mathrm{rel}}(X, t, i) > \\epsilon, \\\\
>     \\mathcal{f}(X, t+1, i) + L1_{\\mathrm{rel}}(X, t, i) & \\text{if } t \\in [0, T-m] \\text{ and } \\mathcal{f}(X, t+1, i) + L1_{\\mathrm{rel}}(X, t, i) \\leq \\epsilon,
> \\end{cases}
> \\tag{1}
> $$
>
> where $\epsilon$ denotes the threshold value and $m$ represents the number of initial timesteps excluded from cache reuse ($m=5$ for MAGI-1, $m=4$ for SkyReels-V2). When $\mathcal{f}(X, t, i) = 0$, forward computation is performed for $i$-th video chunk; otherwise, cached activations are reused. Our empirical analysis demonstrates that excluding early timesteps is critical for preserving generation quality. The threshold values are: MAGI-slow: 0.01, MAGI-fast: 0.015, SkyReels-V2-slow: 0.1, SkyReels-V2-fast: 0.15. We have incorporated the above description into the end of section 3.2 of the paper to help readers understand the details of FlowCache's solution.
>
> **W2: Why the proposed method both out perform the baseline method, TeaCache both in terms of speed and video quality. The reuse operation accelerates the speed, but why it also achieve better frame quality.**
>
> **Reply:**
> Thank you for this insightful observation.
>
> **Quality improvements over the baseline:** The phenomenon of cache reuse improving quality has been observed not only in our work but also in prior literature. For instance, TeaCache [1] reports that both Open-Sora [2] and Latte [3] models achieve performance equal to or exceeding their original counterparts after cache reuse (79.28% vs. 79.22% on Open-Sora; 77.40% vs. 77.40% on Latte). We speculate that cache reuse enables the model to concentrate on specific timesteps, thereby reducing redundancy and enhancing generalization. FlowCache extends this advantage through a more flexible reuse strategy that allows different video frames to focus on independent timestep sets, resulting in superior performance on quality metrics such as video consistency.
>
> **Quality improvements over TeaCache:** As discussed in W1, FlowCache implements independent reuse strategies for each video chunk. Consequently, within the same timestep, different chunks apply customized cache reuse policies tailored to their respective denoising levels. This fine-grained approach further exploits the model's compressible potential, enabling FlowCache to achieve both higher speedup ratios and better video generation quality compared to TeaCache.

---

> > ### Author Response · Authors · 2025-11-27
> > **Response to Reviewer xZAs (Part: 2/2)**
> >
> > **Q1: For the dynamic chunk caching and reuse mechanism, is it adaptively applied across different videos based on the degree of motion dynamics? As mentioned in Weakness 1, the decision-making process remains unclear.**
> >
> > **Reply:** Thank you for this constructive feedback! The specific details of our approach are elaborated in our response to W1 above. Here we provide an important clarification regarding "adaptation based on the degree of motion dynamics."
> >
> > Our current implementation does not support motion-based adaptation at the token level. This is because a video chunk typically comprises multiple frames (e.g., 6 frames in MAGI-1), and tokens within each frame encompass both motion-dynamic pixels (e.g., moving subjects) and relatively static pixels (e.g., background). All tokens within a chunk share the same cache reuse strategy. Currently, FlowCache differentiates reuse schemes solely based on the noise levels of different video chunks in autoregressive video generation models.
> >
> > That said, you raise an intriguing research direction: fine-grained, token-level cache reuse based on motion dynamics. Exploring this would require addressing several key questions:
> > 1. Do tokens with different motion dynamics exhibit significantly different similarity patterns across adjacent timesteps?
> >
> > 2. How can we effectively categorize tokens by their degree of dynamics?
> >
> > 3. Would the computational overhead of such fine-grained control offset its potential benefits?
> >
> > We leave these questions for future investigation. Thank you again for this thought-provoking suggestion!
> >
> > **Q2: Are there any known failure cases for the two proposed designs? For instance, how do they perform on videos with large or complex motion? In such cases, is the speed up improvement less significant?**
> >
> > **Reply:**
> > Thank you for raising this important point.
> >
> > **Failure cases on complex motion:**  We observed that on videos with large or complex motion, Flowcache-fast may exhibit a slight decrease in temporal consistency compared to the original video in the latter frames. However, in contrast, TeaCache suffers from severe degradation in both quality and consistency in these scenarios. Meanwhile, Flowcache-slow maintains excellent consistency and quality, demonstrating robustness.
> >
> > **Speed-up across samples:**  We evaluated our acceleration ratio across diverse samples and found that the speed-up improvement remains stable and consistent across different videos, with little variance between simple and complex motion samples.
> >
> > For better visualization, we have included supplementary video clips in the Appendix. These demonstrations are also available for viewing at the following anonymous GitHub link: https://anonymous.4open.science/r/FlowCache-23495iclrAnonymous/Visualization/VIDEO_COMPARISON.md.
> >
> > **References**
> >
> > [1] Liu et al., Timestep Embedding Tells: It's Time to Cache for Video Diffusion Model. CVPR 2025.
> >
> > [2] Zheng et al., Open-Sora: Democratizing Efficient Video Production for All.
> >
> > [3] Ma et al., Latte: Latent Diffusion Transformer for Video Generation. TMLR 2025.

---

> > > ### Comment · Reviewer_xZAs · 2025-11-28
> > >
> > > Thanks authors for the response. It has addressed all my concerns. I would suggest include the reply of W1 in the final manuscript.

---

> > > > ### Author Response · Authors · 2025-11-29
> > > >
> > > > We sincerely thank the reviewer for the constructive feedback and the time dedicated to reviewing our work.
> > > >
> > > > Regarding your concern in W1, we have explicitly incorporated the relevant content into the revised manuscript. Please refer to Lines 325-337 in the updated PDF.
> > > >
> > > > Best regards!

---

### Author Response · Authors · 2025-11-29
**Summary of Responses to Major Concerns**

We address the major concerns from reviewers regarding FlowCache's mechanism and memory usage.

**Elaboration on Mechanism and Decision Policy**
We use a threshold-based criterion with L1 similarity accumulated **independently** within each video chunk. For the $i$-th video chunk:

$$
\\mathcal{f}(X, t, i) =
\\begin{cases}
    0 & \\text{if } t \\in (T-m, T], \\\\
    0 & \\text{if } t \\in [0, T-m] \\text{ and } \\mathcal{f}(X, t+1, i) + L1 _ {\\mathrm{rel}}(X, t, i) > \\epsilon, \\\\
    \\mathcal{f}(X, t+1, i) + L1 _ {\\mathrm{rel}}(X, t, i) & \\text{if } t \\in [0, T-m] \\text{ and } \\mathcal{f}(X, t+1, i) + L1 _ {\\mathrm{rel}}(X, t, i) \\leq \\epsilon,
\\end{cases}
\\tag{1}
$$

where $\epsilon$ is the threshold and $m$ is the number of excluded initial timesteps ($m=5$ for MAGI-1, $m=4$ for SkyReels-V2). When $\mathcal{f}(X, t, i) = 0$, forward computation is performed; otherwise, cached activations are reused.

**KV Cache Compression Pipeline and Memory Usage**

**1. Pipeline Architecture:**
KV Cache compression begins only after the cache budget is reached. The process operates in two phases:

1.1. *Cache Filling Phase*: New KV cache entries are added until the budget is reached, with no compression occurring during this phase.

1.2. *Compression Phase*: Once the cache is full, the compression mechanism is triggered upon each new KV cache entry arrival to accommodate the new entry.

**2. Memory Optimization:**
Both importance judgments during KV cache compression and input cache retention introduce negligible overhead. For Equation 9, we use only last 50 query tokens. For Equation 11, we derive an equivalent form that reduces memory from 17.79 GB to 0.30 GB and time from 10.26s to 0.077s:
$$
\\frac{1}{L _ k} \\sum _ {i=1}^{L _ k} S^{(h)} _ {ij} =
\\underbrace{\\frac{1}{L _ k} \\sum _ {i=1}^{L _ k} (K _ {\\text{clean}}^{(h)} {K _ {\\text{clean}}^{(h)}}^T) _ {ij}} _ {\\text{Naive: 17.79 GB, 10.26s}}
= \\underbrace{\\left(\\frac{1}{L _ k} \\sum _ {i=1}^{L _ k} (K _ {\\text{clean}}^{(h)}) _ {ij}\\right) {K _ {\\text{clean}}^{(h)}}^T} _ {\\text{Optimized: 0.30 GB, 0.077s}},
\\tag{2}
$$
where $K_{\text{clean}}^{(h)} \in \mathbb{R}^{L_k \times d}$. For the input cache with $L_q = 108000$ and $\text{hidden\\_states}=3072$, the memory footprint is only 0.6 GB.

**3. Memory and Performance Analysis:**
For computational overhead, we profiled FLOPs of token selection and retention in KV cache. For a 10-second 720×720 video with 5-chunk budget, this operation runs only 5 times:

| Query granularity | PFLOPs |
|-------------------|------------|
| Token | 0.0013 |
| Frame | 0.05 |

The actual FLOPs overhead is negligible compared to FlowCache-Fast's total of 140 PFLOPs .

FlowCache addresses memory-quality trade-off through selective compression of global historical information. Under identical memory budgets, our approach outperforms MAGI-1's window attention KV cache strategy (47.65 vs. 45.50). All experiments exclude cache reuse for fair comparison:

| Method | KV Cache Budget (Chunks) | Peak memory (GB) | Physics-IQ |
|--------|---------------------------|-------------------|-----------|
| Vanilla | 8 | 42.84 | 47.60% |
| Vanilla | 5 | 34.26 | 45.50% |
| FlowCache | 7 | 34.05 | 47.55% |
| FlowCache | 6 | 31.25 | 46.72% |
| FlowCache | 5 | 28.45 | 47.65% |

---

### Author Response · Authors · 2025-11-29
**General response (About novelty of our method)**

FlowCache introduces three key innovations for **training-free acceleration of autoregressive video generation**:

1. **Chunkwise Denoising Heterogeneity**: We identify and formalize that denoising progress varies significantly across video chunks—even at the same timestep—necessitating per-chunk caching decisions.

2. **Chunkwise Adaptive Caching**: A novel design where each chunk independently decides whether to reuse or recompute based on its own similarity trajectory.

3. **KV Cache Compression Tailored for Video**: We adapt importance–redundancy scoring to autoregressive video generation KV cache compression by introducing an efficient, equivalence-preserving similarity computation, thereby enhancing cache diversity without sacrificing efficiency.

These contributions collectively make FlowCache the first theoretically grounded, training-free caching framework for efficient autoregressive video generation.

Best regards,
Authors

---

### Meta-Review · Area_Chair_fYRq · 2026-01-07

**Summary:**

The paper presents FlowCache, a novel approach for enhancing inference efficiency in autoregressive video generation models. The method focuses on KV cache compression and selective reuse to improve both computational speed and video generation quality.

While the reviewers raised a few concerns regarding the theoretical details, practical implications, and clarity of the presented methodology, the authors provided substantial clarifications, addressing most of the issues raised. No major unresolved issues remain, and the revisions sufficiently improved the paper's quality. Therefore, I recommend this paper for acceptance.

I urge the authors to incorporate all additional revisions, such as the supplementary video clips and clarifications on memory usage, into the final version to further strengthen the paper's contributions.

**Reviewer Concerns:**

Overall, the authors have effectively addressed the majority of the concerns raised, with no significant issues remaining.

Notably, Reviewer LS46 initially rated the paper 2, raising concerns regarding memory usage, quality comparison, long-video benchmarking, and some questions about the novelty of the approach (such as the window attention mechanism in MAGI-1). In response, the authors:
- Clarified that the additional memory overhead from KV-cache compression is negligible, with experimental and theoretical evidence supporting this.
- Added supplementary video clips, available on GitHub, to showcase superior video quality and speedup ratios.
- Confirmed that VBench-long was used for long-video generation evaluation and updated the manuscript accordingly.
- Explained that KV-cache compression addresses memory efficiency for extended KV ranges, improving model performance without sacrificing physical realism.

**Reviewer Scores:**

As all major concerns have been successfully resolved, reviewer LS46 would likely increase the rating, and other reviewers would maintain their original positive ratings.

---

### Decision · Program_Chairs · 2026-01-26

Accept (Poster)